# Identification of Novel Rotihibin Analogues in *Streptomyces scabies*, Including Discovery of Its Biosynthetic Gene Cluster

Sören Planckaert,[a] Benoit Deflandre,[b] Anne-Mare de Vries,[c] Maarten Ameye,[d] José C. Martins,[c] Kris Audenaert,[d] Sébastien Rigali,[b] Bart Devreese[a]

aLaboratory for Microbiology, Department of Biochemistry and Microbiology, Ghent University, Ghent, Belgium
bInBioS-Centre for Protein Engineering, Institut de Chimie B6a, University of Liège, Liège, Belgium
cNMR and Structure Analysis Group, Ghent University, Ghent, Belgium
dLaboratory of Applied Mycology and Phenomics, Department of Plants and Crops, Ghent University, Ghent, Belgium

**ABSTRACT** *Streptomyces scabies* is a phytopathogen associated with common scab disease. This is mainly attributed to its ability to produce the phytotoxin thaxtomin A, the biosynthesis of which is triggered by cellobiose. During a survey of other metabolites released in the presence of cellobiose, we discovered additional compounds in the thaxtomin-containing extract from *Streptomyces scabies*. Structural analysis by mass spectrometry (MS) and nuclear magnetic resonance (NMR) revealed that these compounds are amino acid sequence variants of the TOR (target of rapamycin) kinase (TORK) pathway-inhibitory lipopeptide rotihibin A, and the main compounds were named rotihibins C and D. In contrast to thaxtomin, the production of rotihibins C and D was also elicited in the presence of glucose, indicating different regulation of their biosynthesis. Through a combination of shotgun and targeted proteomics, the putative rotihibin biosynthetic gene cluster *rth* was identified in the publicly available genome of *S. scabies* 87-22. This cluster spans 33 kbp and encodes 2 different nonribosomal peptide synthetases (NRPSs) and 12 additional enzymes. Homologous *rth* biosynthetic gene clusters were found in other publicly available and complete actinomycete genomes. Rotihibins C and D display herbicidal activity against *Lemna minor* and *Arabidopsis thaliana* at low concentrations, shown by monitoring the effects on growth and the maximal photochemistry efficiency of photosystem II.

**IMPORTANCE** Rotihibins A and B are plant growth inhibitors acting on the TORK pathway. We report the isolation and characterization of new sequence analogues of rotihibin from *Streptomyces scabies*, a major cause of common scab in potato and other tuber and root vegetables. By combining proteomics data with genomic analysis, we found a cryptic biosynthetic gene cluster coding for enzyme machinery capable of rotihibin production. This work may lead to the biotechnological production of variants of this lipopeptide to investigate the exact mechanism by which it can target the plant TORK pathway in *Arabidopsis thaliana*. In addition, bioinformatics revealed the existence of other variants in plant-associated *Streptomyces* strains, both pathogenic and nonpathogenic species, raising new questions about the actual function of this lipopeptide. The discovery of a module in the nonribosomal peptide synthetase (NRPS) that incorporates the unusual citrulline residue may improve the prediction of peptides encoded by cryptic NRPS gene clusters.

**KEYWORDS** *Streptomyces*, TORK, common scab, lipopeptide, nonribosomal peptide, proteomics

Address correspondence to Bart Devreese, bart.devreese@ugent.be.

Discovery of the rotihibin biosynthetic cluster

*S*treptomyces scabies (syn. *S. scabiei*) is a plant-pathogenic bacterium causing common scab disease, resulting in substantial damage to potatoes and other root and tuber crops, including carrot, radish, beet, parsnip, and turnip. Common scab disease is widely distributed and seriously diminishes the market value of the crops (1). The

disease is characterized by deep-pitted and corky lesions on the root or tuber surface (2). Other *Streptomyces* species, *S. reticuliscabiei*, *S. cheloniumii*, and *S. ipomoeae*, are responsible for netted scab, russet scab, and soil rot of sweet potato, respectively (3–5).

In the last decades, several studies focused on elucidating the molecular mechanisms of virulence of *S. scabies*. The production of thaxtomin A has been shown to be a key pathogenicity determinant (6). The *txtABCDEH* gene cluster is responsible for the biosynthesis of this 4-nitroindol-3-yl-containing 2,5-dioxo-piperazine. These thaxtomin biosynthetic genes are highly conserved across plant-pathogenic streptomycetes and reside on a pathogenicity island that is mobilized in some species (7). Thaxtomin A primarily targets expanding host tissue by affecting cellulose synthase complex density, expression of cell wall genes, and cell wall composition (8, 9).

The production of thaxtomin A is strictly controlled, involving several layers of regulation. Cellobiose, together with cellotriose, is recognized as the main specific elicitor of thaxtomin A biosynthesis in *S. scabies* (10). Once imported via the CebEFG-MsiK ABC transporter (11), these products of cellulose turnover directly target the pathway-specific transcriptional activator TxtR and the cellulose utilization regulator CebR, which together constitute a double-locking system on the *txtABCDE* gene cluster. Specific interaction between CebR and cellobiose triggers the release of the repressor from different binding sites within the thaxtomin biosynthetic gene cluster (BGC), including *txtR*. This results in the transcriptional activation of *txtA* and *txtB*, consequently inducing thaxtomin A production and pathogenicity (12, 13). In addition, several *bld* global regulators directing secondary metabolism or morphological differentiation are involved in the regulation of thaxtomin synthesis (14).

It is believed that plant-pathogenic streptomycetes produce other important phytotoxins involved in pathogenicity. Using proteomics, we previously demonstrated that the levels of enzymes involved in the biosynthesis of other secondary metabolites like concanamycin A and coronafacoyl phytotoxins are also dependent on cellobiose levels, a finding later confirmed by measuring altered levels of these compounds in the presence of cellobiose and/or upon *cebR* deletion (15). Natsume et al. isolated concanamycins A and B from Japanese *S. scabies* isolates (16). Recently, they demonstrated root growth-inhibitory activity, necrosis-inducing activity, and a synergistic effect with thaxtomin A (17). Coronafacoyl phytotoxins contribute to the development of root disease symptoms and cause hypertrophy of potato tuber tissue (18). The causative agent of russet scab, *S. cheloniumii*, produces FD-891, which induces necrosis of potato tuber tissue (19). All these data indicate that multiple secondary metabolites are involved in scab disease. Moreover, thaxtomin A-deficient streptomycetes that are also able to cause scab disease have been isolated. *Streptomyces* sp. strain GK18 produces borrelidin, which is reported to cause severe deep, black holes on potato tuber slices (20, 21). Similarly, fridamycin E was isolated from an *S. turgidiscabies* strain from a netted scab lesion in Sweden. This phytotoxin was demonstrated to reduce or even inhibit the sprouting of *in vitro* microtubers (22).

Driven by the discovery that under thaxtomin production-inducing conditions, i.e., the addition of cellobiose, multiple proteins potentially involved in secondary metabolism were increased in abundance, we further analyzed extracts from the extracellular medium of *Streptomyces scabies* RL-34 in order to obtain insight into the metabolites secreted by this organism when grown in the presence of cellobiose. We report here the discovery of two new compounds that were characterized to be variants of the lipopeptide rotihibins A and B, previously discovered by Fukuchi et al. in extracellular extracts of *Streptomyces* sp. strain 3C02, later designated an *S. graminofaciens* strain (23, 24). A culture filtrate containing rotihibins A and B inhibits the growth of lettuce seedlings, while purified rotihibin A causes shoot stunting in tobacco seedlings at low concentrations (25). Recently, it was demonstrated that rotihibin A acts as a TOR (target of rapamycin) kinase (TORK) pathway inhibitor (26). Therefore, we assessed the plant growth-inhibitory effect of the newly isolated compounds, designated rotihibins C and D, and found a severe effect on growth and photosystem II photochemistry efficiency

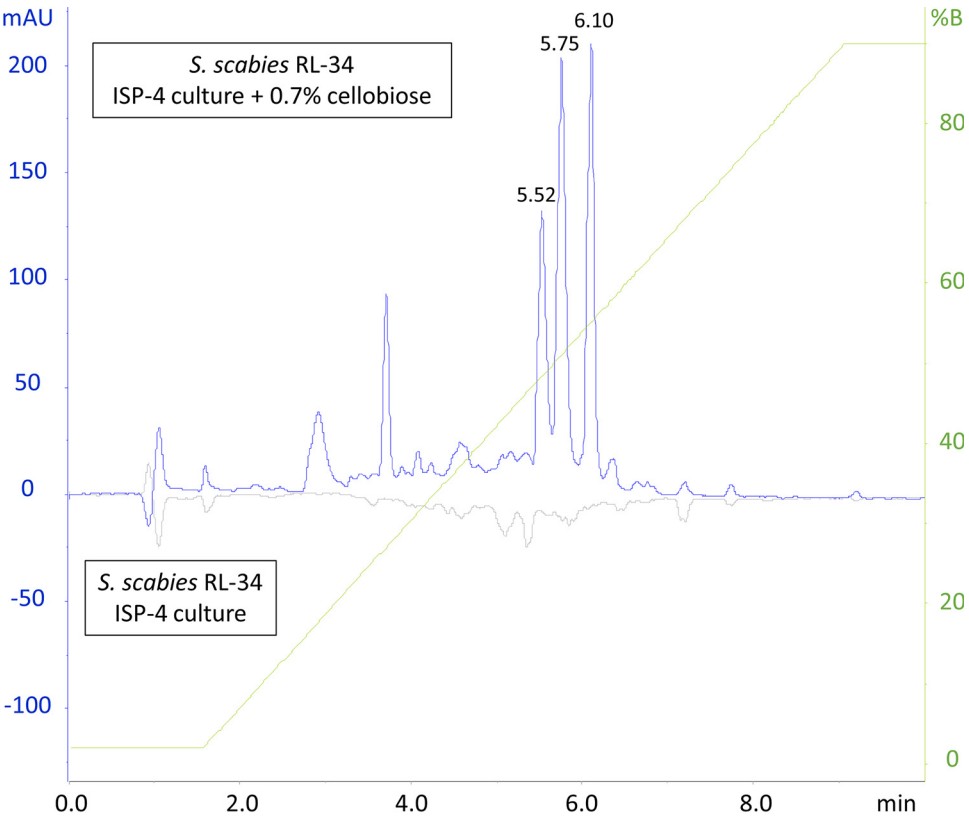

**FIG 1** Effect of cellobiose on the secreted metabolome of *Streptomyces scabies* RL-34. HPLC runs were performed to compare *n*-butanol extracts of *Streptomyces scabies* RL-34 ISP-4 cultures grown in the presence of cellobiose (upward chromatogram) to those of cultures grown in the absence of cellobiose (downward chromatogram). mAU, milli-absorbance unit.

in *Lemna minor* L. and *Arabidopsis thaliana* L. Heynh. Furthermore, we provide data that demonstrate that the biosynthetic machinery to produce rotihibins C and D is encoded by a gene cluster covering a 33-kb segment containing 14 open reading frames (ORFs) that is conserved in both pathogenic and nonpathogenic plant-associated *Streptomyces* species.

## RESULTS AND DISCUSSION

**Rotihibin production by *Streptomyces scabies*.** Cellobiose is known to induce thaxtomin A production in *Streptomyces scabies* (10). We postulated that cellobiose also elicits the production of other virulence factors important for infection of root and tuber crops. When comparing the high-performance liquid chromatography (HPLC) profiles of *n*-butanol extracts of media from cultures obtained from *S. scabies* RL-34 grown in International *Streptomyces* Project medium 4 (ISP-4) either supplemented or not with 0.7% cellobiose, new peaks appeared in the HPLC profile of the extract obtained after growth in the presence of cellobiose (Fig. 1). We focused further analysis on the peaks eluting at 5.52, 5.75, and 6.10 min, and the corresponding metabolites were characterized by high-resolution electrospray ionization mass spectrometry (HR-ESI-MS). A distinct ion peak at *m/z* 439.16 [M + H]$^+$ could be identified in the spectrum obtained from the fraction eluting at 6.10 min, accompanied by peaks at *m/z* 421.15 [M − H$_2$O + H]$^+$, 461.14 [M + Na]$^+$, and 477.11 [M + K]$^+$ (data not shown). This pattern of *m/z* values can be attributed to thaxtomin A (theoretical monoisotopic mass, 438.11), confirming that our culture conditions induced the production of the main pathogenic determinant of *S. scabies*.

The ESI spectrum of the fraction eluting at 5.52 min displayed a major peak at *m/z* 860.47, while the spectrum of the fraction eluting at 5.75 min displayed a peak at *m/z* 874.47, which is also present in the 5.52-min sample (see Fig. S1 in the supplemental

material). Both molecular ions were selected for tandem MS (MS/MS) analysis. Putative daughter $b$-ions at $m/z$ 742 [M + H − 118]$^+$, 612 [M + H − 118 − 130]$^+$, and 310 [M + H − 118 − 130 − 302]$^+$ and $y$-ions at $m/z$ 551 [M + H − 309]$^+$, 450 [M + H − 309 − 101]$^+$, and 249 [M + H − 309 − 101 − 201]$^+$ were observed in ESI-MS/MS analyses of the precursor ion at $m/z$ 860.47 (Fig. 2). The bioinformatics tool Insilico Peptidic Natural Products Dereplicator was used to dereplicate the structure through database searching of mass spectra (27). The mass spectrum of rotihibin A, a nonribosomal peptide (NRP) functional as a plant growth regulator in *Streptomyces graminofaciens*, was found to have the best fit to our data. However, the spectrum of our newly detected compound from *S. scabies* indicated the presence of a threonine in the NRP backbone instead of a serine (as in rotihibin A) (Fig. 2A) (25). Indeed, the neutral losses of 118, 130, 302, 309, 101, and 201 mass units correspond to the masses of asparaginol (Asn-ol); hydroxy-asparagine (OH-Asn); (*allo*)-threonine (aThr), 2,4-diamino butyric acid (Dab), and threonine (Thr); 2-*cis*-decenoic acid (*cis*-DA) and citrulline (Cit); threonine (Thr); and aThr and 2,4-diamino butyric acid, respectively (Fig. 2B). Previously, Halder et al. established a chemical synthesis pathway of rotihibin A and structural analogues (26). In their structural analogue RotA-D3, the serine residue at position 2 of the amino acid chain was replaced by alanine. This derivative turned out to be more active than rotihibin A, as shown by the overall more pronounced plant growth retardation in an *A. thaliana* bioassay (26). Interestingly, our data suggest that *S. scabies* RL-34 would produce a natural rotihibin variant that is modified at the same position.

Fragmentation of the ion at $m/z$ 874.47 displayed the same fragment ions, except for the daughter ions containing the acyl chain ($m/z$ +14), which could be explained by a longer/branched acyl chain. For further description, we named the compound at $m/z$ 860 rotihibin C and the compound at $m/z$ 874 rotihibin D. For each of the compounds, we also see a component that is 2 Da smaller, which we assume to be due to desaturation. We also see a minor component at $m/z$ 844/846, which, according to the MS/MS spectrum (data not shown), is a similar compound with a shorter fatty acyl chain.

Glucose is known to suppress thaxtomin A production in *Streptomyces scabies* and *Streptomyces acidiscabies* (28, 29). The effect of glucose, a breakdown product of cellobiose, on the production of the rotihibin analogues was tested. Indeed, the peak eluting at 6.1 min, corresponding to thaxtomin A, disappeared in the HPLC profile of the *S. scabies* RL-34 cultures grown in the presence of glucose. In contrast, compounds eluting at 5.52 and 5.75 min were still present (Fig. 3), and they were confirmed by direct-infusion HR-ESI-MS/MS to be rotihibins C and D (data not shown). This is a clear indication that cellobiose is not the trigger for rotihibin production in *S. scabies* RL-34 as it is for thaxtomin A.

Finally, we verified whether rotihibins C and D are also produced by other *S. scabies* strains. We performed similar extraction and purification methods on extracellular medium from the hypervirulent *S. scabies* 87-22 strain (12). The chromatograms and ESI spectra of these components were identical, indicating that this strain is also able to produce the same compounds (data not shown).

To confirm the structure proposed from the MS data, HPLC-purified rotihibins C and D were characterized using $^1$H nuclear magnetic resonance (NMR). Superposition of the two-dimensional (2D) total correlation spectroscopy (TOCSY) spectra of both compounds as depicted in Fig. 4 clearly indicates the presence of identical correlation patterns. In addition, analysis of the various spin systems in the TOCSY spectrum showed good agreement with the amino acid residues expected from the MS/MS analysis of the peptide chains of rotihibins C and D. Specifically, it agrees with our suggestion that a threonine in rotihibin C replaces the serine observed in rotihibin A. Since the NMR spectra of purified rotihibins C and D did not reveal structural differences between the peptide chains, the NMR analysis also supports that the 14-Da difference in mass likely results from an extra methylene group for the acyl chain in rotihibin D; however, the spectral overlap between the many CH$_2$ units does not allow us to clearly establish acyl chain length using NMR.

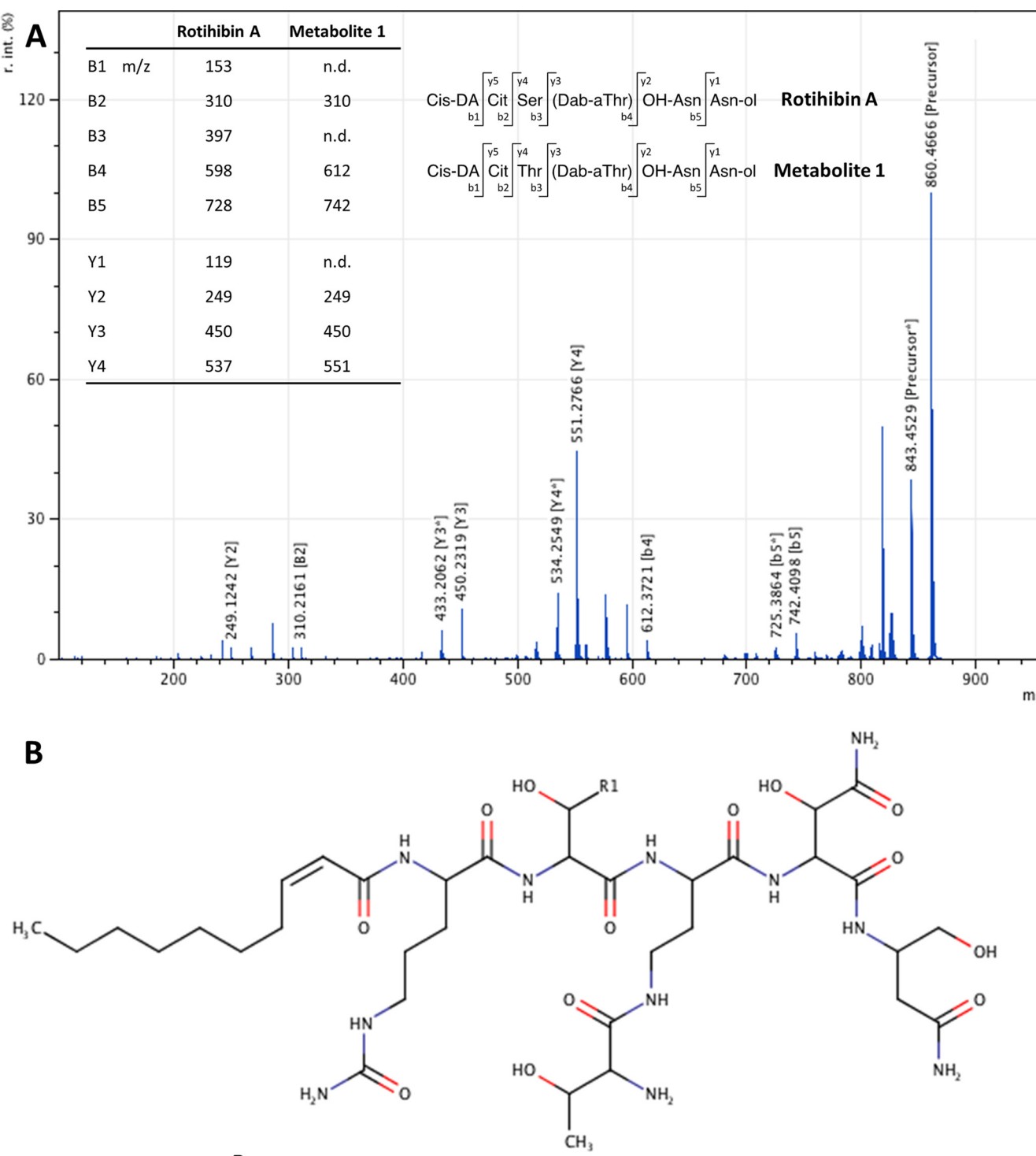

**FIG 2** (A) MS/MS spectrum of the metabolite eluting after 5.52 min as displayed in Fig. 1. The fragment ions of this compound, designated rotihibin C (metabolite 1), are compared with those from rotihibin A, displaying a difference only in the NRP backbone (Ser→Thr). n.d., not detected. (B) Proposed structure of rotihibin C compared to the previously identified rotihibin A.

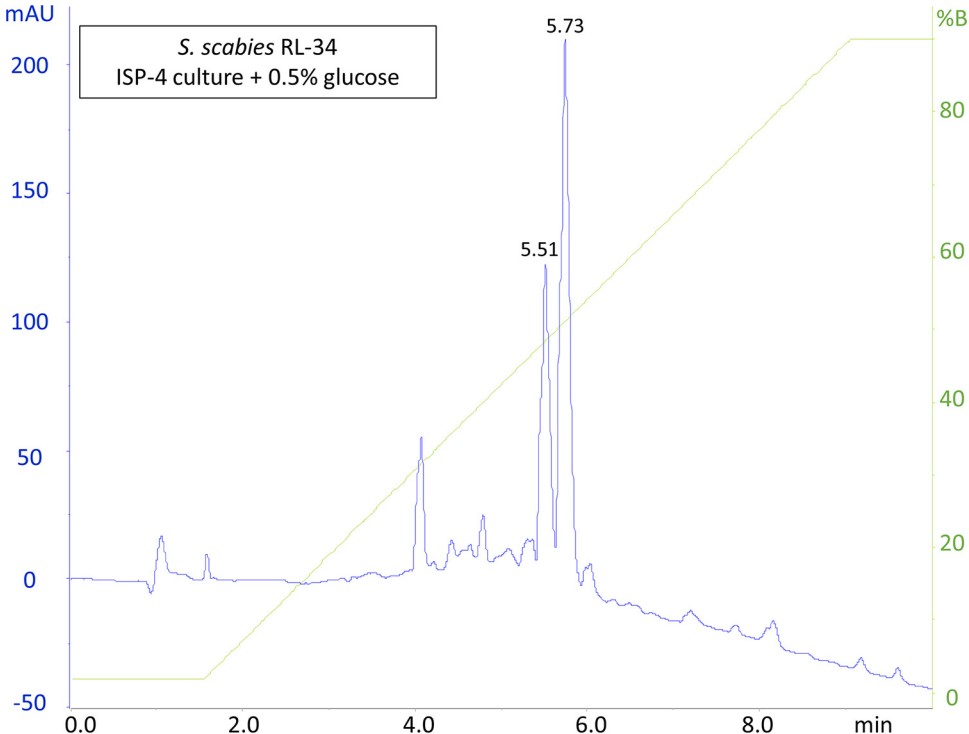

**FIG 3** Effect of glucose on the secreted metabolome of *Streptomyces scabies* RL-34. An HPLC run was performed on *n*-butanol extracts of *Streptomyces scabies* RL-34 ISP-4 cultures grown in the presence of glucose. mAU, milli-absorbance unit.

**Identification of the biosynthetic gene cluster responsible for rotihibin production.** Apart from the thaxtomin biosynthetic gene cluster (BGC), *S. scabies* also possesses four other NRP synthase (NRPS)-type BGCs; two of them are cryptic, that is, BGCs for which no biomolecule has yet been associated with the genetic material. Proteins associated with a cryptic NRPS-type BGC (from *scab_3221* to *scab_3351*) (see details below) were found in a previous proteomic experiment (15). We investigated the possible relationship between this BGC and rotihibin production by targeted proteomics using multiple-reaction monitoring (MRM). The proteotypic peptides of SCAB_3241 (LIDEEPYR and ATGLSDEEFLAR), SCAB_3251 (IPVYLAALGPK and IDVGSAVLQIPAR), SCAB_3281 (VTDEQLAALDLSR and EDPLLTDALAGQR), SCAB_3291 (GQLPEGAWR and LGTADLWLR), SCAB_3301 (LYGGAATDIPHVR and SELAGVFADLLR), and SCAB_3321 (AWIDSDLATPVPVTGER, ADTSGDPTFEELLDR, and GGTVPFAVPAALR) were used to evaluate the protein levels. Figure 5 shows the positive correlation between the presence of either glucose or cellobiose and the production of the selected proteins, thereby providing the first evidence that rotihibin production and this cryptic BGC might be linked. Interestingly, the inactivation of the cellulose utilization regulator *cebR* had no effect on the production of enzymes for this cluster or even instead had the reverse effect for the *scab_3221* gene product (15). This indicates that the effect of cellobiose on the expression of the NRPS gene cluster is not dependent on *cebR* and that there is no coregulation with thaxtomin production.

We then performed a bioinformatic analysis of this gene cluster to investigate whether it effectively contains the information for enzyme machinery allowing rotihibin production. We first analyzed the cryptic gene cluster using antiSMASH (30). It resides on a 70.1-kb segment of the *S. scabies* genome with 51 open reading frames (ORFs). The region around the largest NRPS gene (*scab_3321*) was compared with all public genomes using PATRIC (31). This analysis confined the cluster to a 33-kb segment with 14 ORFs (*scab_3221* to *scab_3351*). The presence of a putative integrin protein (SCAB_3201) and a putative transposase (SCAB_3371) at the borders of this cluster is evidence for the possible mobilization of this gene cluster.

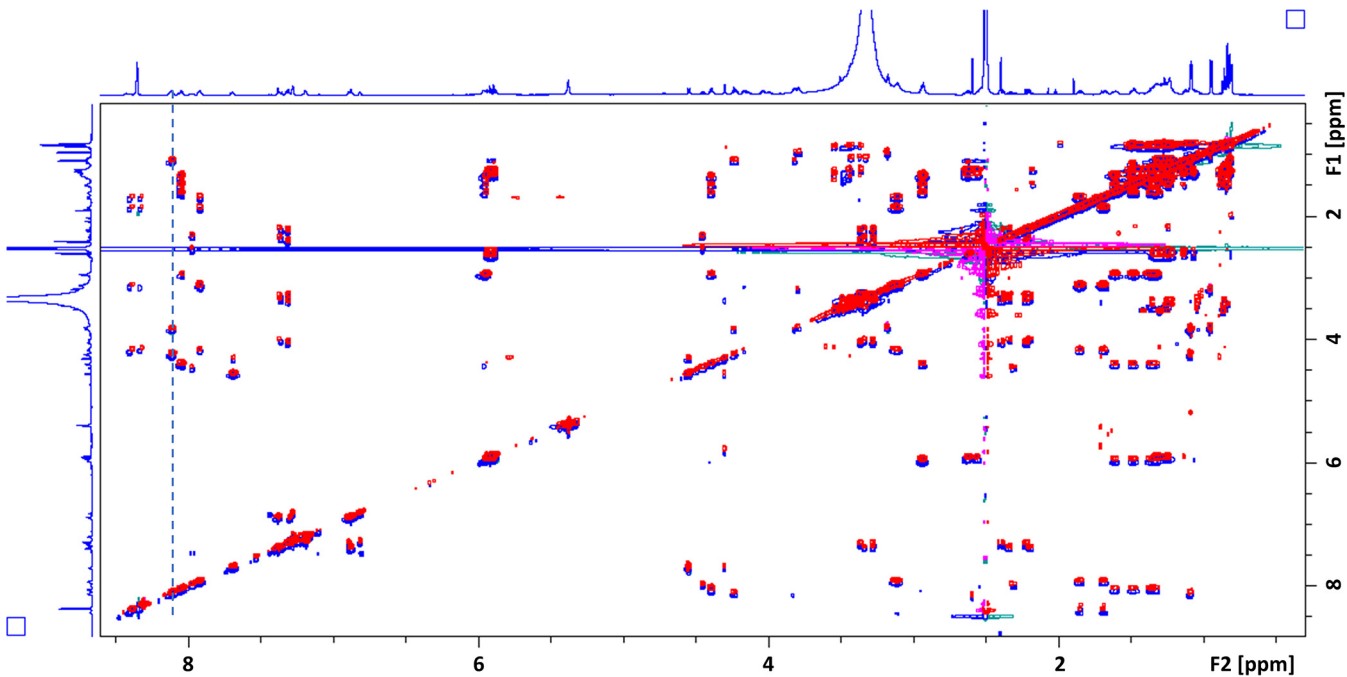

**FIG 4** 2D ¹H-¹H TOCSY spectra used for the analysis of the rotihibin C and D structures. The 2D TOCSY spectra of rotihibins C (blue) and D (red) are shown overlaid, with a slight offset of one with respect to the other to clearly show the identity of the correlation patterns for the peptide chains in both compounds. The dashed line identifies the correlation belonging to the threonine residue.

The sequences of these 14 ORFs were individually submitted to a BLAST search against the UniProtKB/Swiss-Prot database, and putative functions derived from this search are listed in Table 1 and Fig. 6. A preliminary analysis of these functions allowed us to conclude that this gene cluster contains the necessary information to produce the enzymes needed for the biosynthesis of rotihibins C and D, as outlined below. Therefore, this cluster is further referred to as the rotihibin (*rth*) gene cluster.

**Analysis of the genes involved in the rotihibin BGC. (i) NRPS genes for peptide chain assembly.** Two NRPSs, RthA and RthB, are present in this cryptic gene cluster. RthA is predicted to contain 5 modules, while RthB contains a single module for the incorporation of amino acids in a peptide chain (Fig. 7A). Different tools were used to predict their NRPS adenylation domain (A domain) specificity (Table 2), and from these results, we propose a biosynthetic path for the production of the rotihibin peptide chain (Fig. 7B). Module 1 of RthA was predicted to be specific for Glu or Ser, depending on the tool used. However, we believe that module 1 is responsible for the incorporation of citrulline, the first amino acid of the peptide: the signature sequence for an A domain that could be specific for citrulline is possibly not included in the prediction tools. The A domain was missing from module 2, while (*allo*)-threonine should be built into the rotihibin backbone. RthB is the best candidate to introduce threonine into the rotihibin peptide sequence, as its A domain is predicted to be specific for threonine. This phenomenon was previously described in the enduracin biosynthetic gene cluster from *Streptomyces fungicidicus*. Another NRPS activates and transfers L-*allo*-threonine to the module with the missing A domain (32). RthD, a type II thioesterase-like (TEII) enzyme, is believed to assist in the shuttling of the activated aThr between the stand-alone A-T didomain module RthB and the A-less C-T module of RthA, as described previously for WS9326A biosynthesis (33). A serine residue was predicted in module 3, but (2,4)-di-amino butyric acid was observed. The prediction of the two last modules to introduce asparagine residues is in line with the rotihibin structure having two asparagine-like residues at its C terminus. Module 5 has a reductase (RE) domain. This domain is known to reduce the peptidyl thioester into its corresponding alcohol, which explains the presence of L-asparaginol (34). This also explains why rotihibin does not form a cyclic structure like many other NRPS-derived lipopeptides.

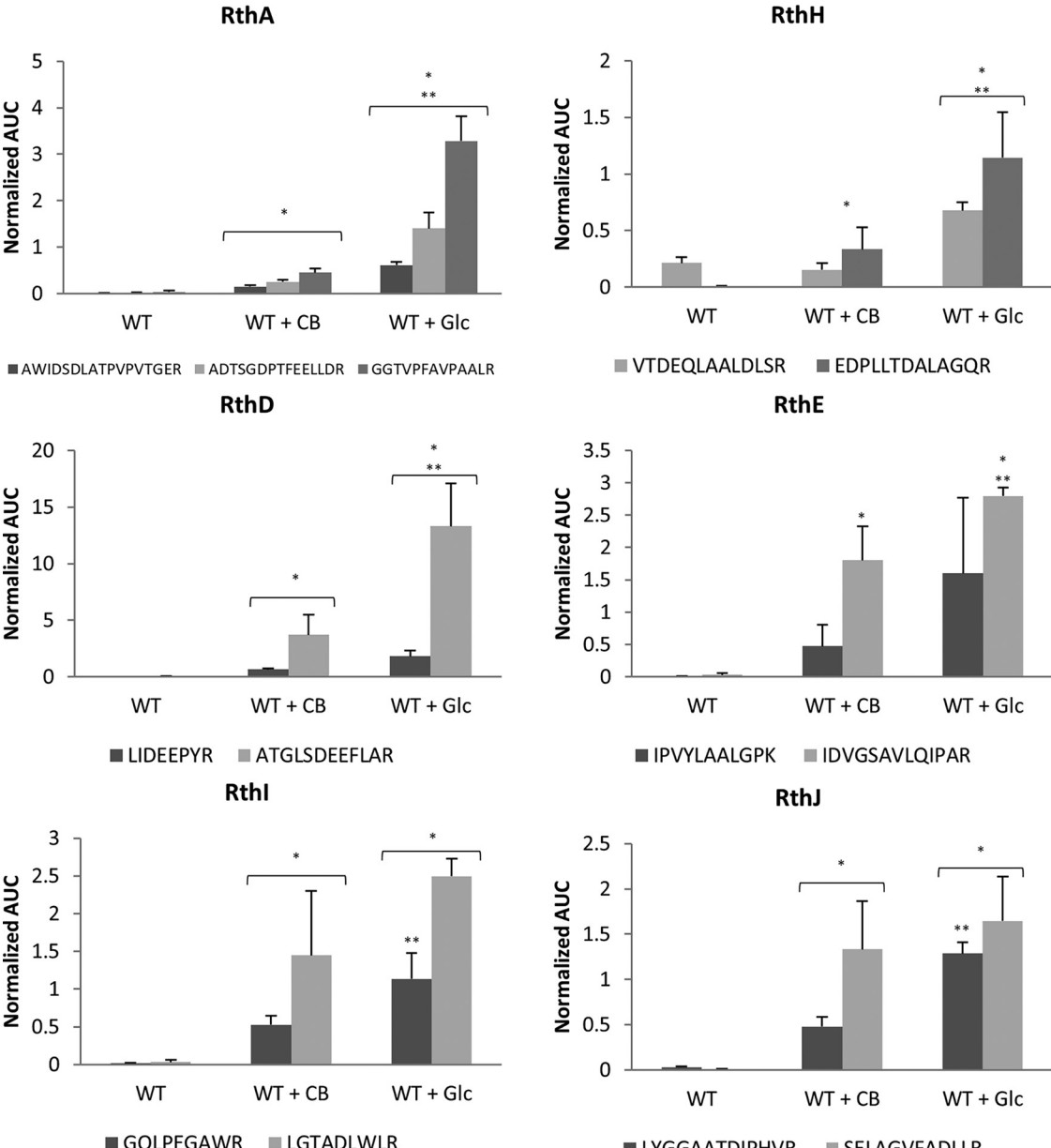

**FIG 5** Relative abundances of proteins as part of nonribosomal peptide machinery in response to cellobiose (CB) (Glc$_2$) or glucose (Glc) supply determined by targeted proteomics (LC-MRM). The plot displays the average area under the curve for each peptide used as a marker for the different proteins, normalized to the spike-in standard BSA. These results show significant normalized quantitative peptide abundances ($P < 0.05$) compared to the wild-type (WT) strain grown in ISP-4 without supplementary oligosaccharides (*) and with cellobiose (**). A statistical two-sided Student $t$ test (homoscedastic) was performed. The error bars plot the standard deviations (SD) from three biological replicates. Note that we already adopted the rotihibin gene cluster annotation (see the text and Table 1).

Potentially, RthB has a dual function by transferring ʟ-*allo*-threonine to module 2 and building an additional ʟ-*allo*-threonine into the NRP backbone by forming an *iso*-peptide bond with (2,4)-di-amino butyric acid. This was also described for viomycin and capreomycin, where one A domain is proposed to function twice, to acetylate not only the T domain within its own module but also the T domain of another module (35, 36).

The stereochemistry of the different amino acids of rotihibin A was previously determined by Marfey's method (23). 2,4-Diamino butyric acid (Dab), aThr, Asn-ol, and hydroxy-asparagine (OH-Asn) were proven to be in the ʟ-form, and the citrulline (Cit) residue was proven to be in the ᴅ-form. The epimerization (E) domain in module 1 confirms the results of this

**TABLE 1** Functions of proteins encoded by the rotihibin biosynthetic gene cluster

| Gene | Protein | Predicted function |
|---|---|---|
| SCAB_3221 | RthB | Unimodular nonribosomal peptide synthetase |
| SCAB_3231 | RthC | Short-chain dehydrogenase/reductase |
| SCAB_3241 | RthD | Type II-like thioesterase enzyme |
| SCAB_3251 | RthE | $F_{420}$-dependent oxidoreductase |
| SCAB_3261 | RthF | ABC transporter permease |
| SCAB_3271 | RthG | ABC transporter ATP-binding protein |
| SCAB_3281 | RthH | Fatty acyl-AMP ligase |
| SCAB_3291 | RthI | Acyl-CoA dehydrogenase |
| SCAB_3301 | RthJ | Acyl-CoA dehydrogenase |
| SCAB_3311 | RthK | Acyl carrier protein |
| SCAB_3321 | RthA | Pentamodular nonribosomal peptide synthetase |
| SCAB_3331 | RthL | MbtH-like protein |
| SCAB_3341 | RthM | L-Asparagine oxygenase |
| SCAB_3351 | RthN | Diaminobutyrate-2-oxoglutarate transaminase |

experiment, as it is known to convert amino acids between the L- and D-isomers. Many NRPSs require MbtH-like proteins (MLPs) for the proper folding and activity of the NRPS. This protein is essential for the biosynthesis of, for example, pyoverdine (37), vancomycin (38), coelichelin, and the calcium-dependent antibiotic (CDA) (39). *rthL*, encoding an MbtH-like protein, is probably necessary for the production of the rotihibins. Li et al. showed that RthL can functionally replace TxtH in the thaxtomin biosynthetic pathway, demonstrating that MLPs from different pathways are able to complement each other (40).

**(ii) Biosynthesis of nonproteinogenic amino acids.** Three nonproteinogenic amino acids are incorporated into the rotihibin nonribosomal peptide backbone: Cit, Dab, and OH-Asn. *rthM* encodes an L-asparagine oxygenase (AsnO), which we propose to be involved in converting Asn to OH-Asn. AsnO was previously found in the biosynthetic gene cluster of CDA (41) and A54145 (42). Finally, the unusual amino acid 2,4-diamino butyric acid is produced from aspartate $\beta$-semialdehyde by the enzyme diaminobutyrate-2-oxoglutarate transaminase (43, 44). *rthN* encodes a homologue of this enzyme and is proposed to be responsible for Dab biosynthesis, which is finally incorporated into rotihibin. There is no specific gene that could be responsible for citrulline synthesis, but this is an intermediate of arginine biosynthesis that could be sufficiently available.

**(iii) Formation and attachment of the fatty acid tail.** Numerous lipopeptides, synthesized by NRPS complexes, have already been described in *Streptomyces* species, for example, A54145 (42), CDA (45), daptomycin (46), and enduracidin (32). These clusters

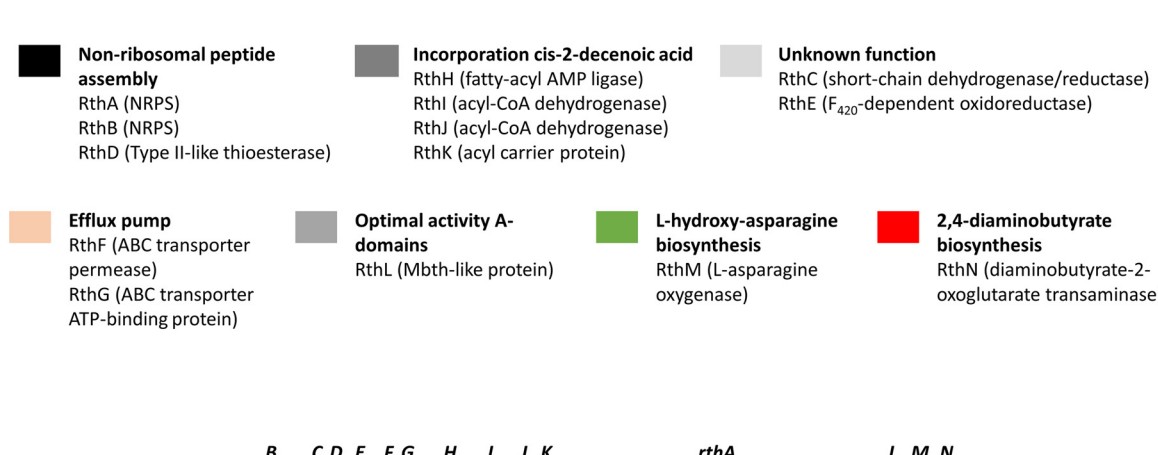

**FIG 6** The rotihibin gene cluster proposed to be responsible for the production and secretion of rotihibins.

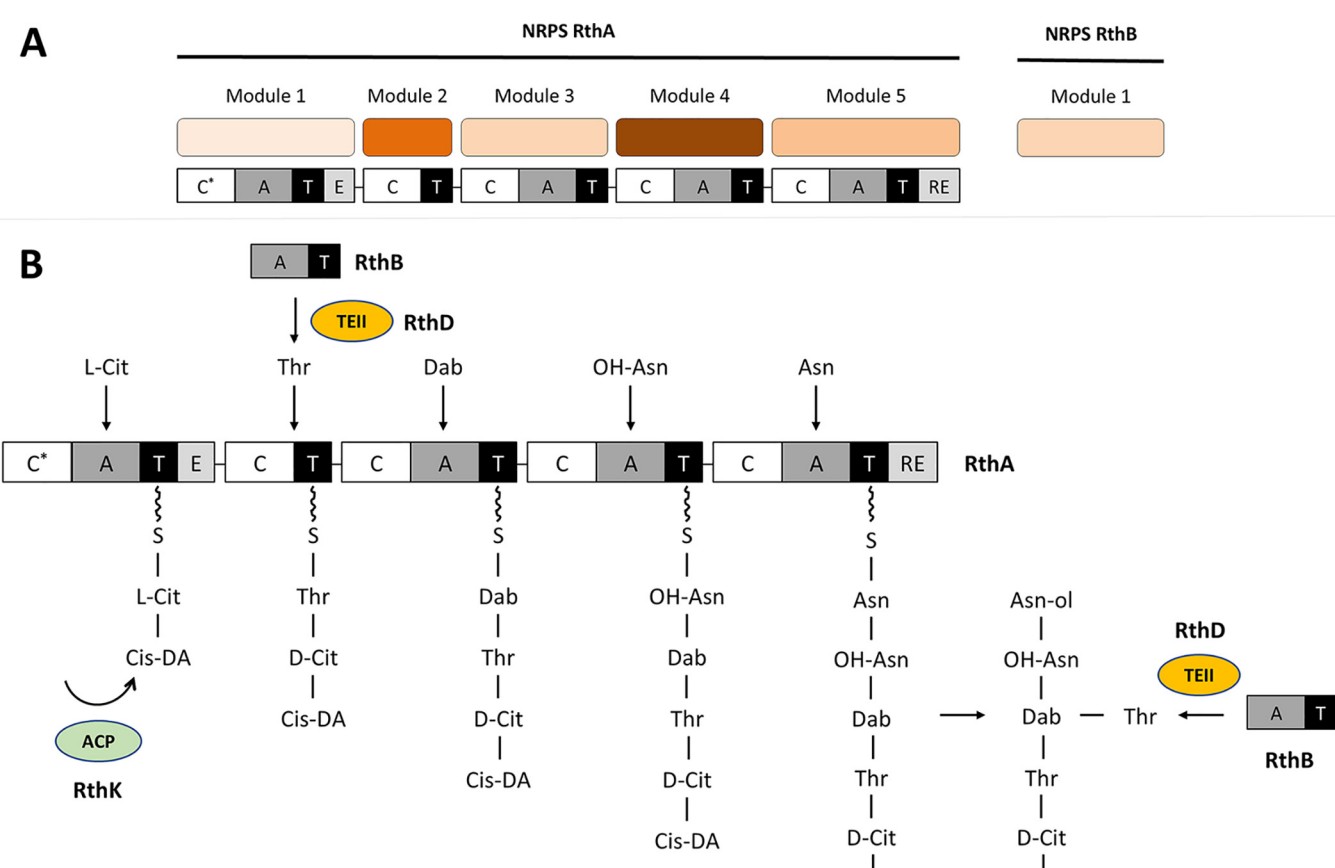

**FIG 7** (A) Organization of the NRPSs in the rotihibin gene cluster. RthA contains five modules, while RthB has only one module. C, condensation; A, adenylation; T, thiolation; E, epimerization; RE, reduction. (B) Proposed biosynthetic pathway and modular organization of the NRPS for rotihibin biosynthesis. RthA is responsible for the NRP backbone. Cis-DA, *cis*-2-decenoic acid; Cit, citrulline; Dab, (2,4)-di-amino butyric acid; C, condensation domain; A, adenylation domain; T, thiolation; E, epimerization domain; RE, reductase domain; ACP, acyl carrier protein.

contain genes whose products act together to acylate the first amino acid. Bioinformatic analysis of the *rth* gene cluster revealed the presence of a gene encoding an acyl carrier protein (*rthK*), which is immediately flanking *rthA*. This protein is proposed to transfer medium-chain fatty acids to the N-terminal domain of the Cit-incorporating module (Fig. 8). Fatty acyl-AMP ligases are enzymes establishing the cross talk between fatty acid synthases and NRPSs or polyketide synthetases (PKSs). They initiate the biosynthesis of lipopeptides by the activation of a fatty acyl residue and occur with a high incidence in putative lipopeptide NRPS/PKS clusters

**TABLE 2** NRPS adenylation domain specificity prediction[a]

| | Domain specificity predicted by method | | | | | | | | |
|---|---|---|---|---|---|---|---|---|---|
| | LSI based | | NRPSsp | | PKS/NRPS | | SEQL-NRPS | | Corresponding amino acid |
| Module | Pred | Score (%) | Pred | Score (HMMER bit) | Pred | Score (bits) | Pred | Prob | |
| RthA | | | | | | | | | |
| 1 | Glu | 0.595 | Ser | 661.3 | Glu | 16.5 | Ser | 0.485 | Lipo-D-Cit |
| 2 | | | | | | | | | L-*allo*-Thr |
| 3 | Arg | 0.528 | Ser | 568.5 | Ser | 16.9 | Ser | 0.474 | L-Dab |
| 4 | Asn | 0.934 | Asn | 486.6 | Asn | 16.5 | Asn | 0.615 | L-Asn-OH |
| 5 | Asn | 0.700 | Asn | 450.4 | Asn | 18.1 | Orn | 0.463 | L-Asn |
| RthB | | | | | | | | | |
| 1 | Thr | 0.955 | Thr | 651.9 | Thr | 19.2 | Thr | 0.469 | Thr |

[a]The specificities of the A domains of the different modules in RthA and RthB were predicted using different software tools. Pred, predicted amino acid; Prob, probability.

**FIG 8** Proposed activation and transfer of *cis*-2-decenoic acid (*cis*-DA).

(47). RthH is supposed to recruit a fatty acid and transfer it to the acyl carrier protein RthK. The fatty acid side chain lengths and/or degrees of saturation are the only differences between the rotihibin C and D analogues. The desaturation could be explained by the presence of two acyl-CoA dehydrogenases (RthI and RthJ), similar to the situation in the ramoplanin biosynthetic gene cluster (48). One is expected to introduce the first double bond, while the second additional dehydrogenation is possibly not essential. The minor amounts of other lipopeptides identified via MS, which differ by only $-2$ Da, could be explained this way (Fig. S1). This phenomenon was previously described in acyl-desferrioxamines of *Streptomyces coelicolor* (49).

**(iv) Self-resistance genes.** During antibiotic production, transporter and transporter-associated proteins are important for the import of effector molecules, self-resistance, and guiding/exporting the antibiotic to the extracellular environment. *rthF* and *rthG* are predicted to act as the rotihibin-exporting machinery as they encode an ABC-type permease and an ABC transporter ATP-binding protein, respectively, a combination often found in antibiotic biosynthetic gene clusters. RthF shows 47% similarity with the daunorubicin/doxorubicin resistance ABC transporter permease protein DrrB, while RthG shows 61% similarity with the daunorubicin/doxorubicin resistance ATP-binding protein DrrA in *Streptomyces peucetius*. This DrrAB efflux system in *S. peucetius* has been shown to be a multidrug transporter with broad specificity (50). RthFG possibly has a similar role in the self-resistance mechanism of *S. scabies*.

**(v) Other genes within the *rth* cluster.** The BGC for rotihibin in *S. scabies* houses a number of genes not directly associated with rotihibin assembly. The protein product of *rthE* is predicted to be an $F_{420}$-dependent oxidoreductase, and *rthC* encodes a short-chain dehydrogenase/reductase. These two enzymes are, for example, directly involved in the production of coronafacoyl phytotoxins in *S. scabies* (51). The impact of the inactivation of these genes on rotihibin biosynthesis has to be studied to reveal their exact role.

**(vi) Inactivation of *rthB* abolishes rotihibin production.** To ascertain whether the *rth* cluster is responsible for the production of rotihibins, the *scab_3221* gene was disrupted and replaced in *S. scabies* 87-22 by an apramycin resistance cassette (11, 13, 29). HPLC analysis and targeted metabolomics on *n*-butanol extracts of the Δ*rthB* strain confirmed the absence of rotihibins (Fig. 9). These data support the evidence that the *rth* BGC is indeed responsible for rotihibin production. Simple complementation of the Δ*rthB* strain with an integrative plasmid harboring an intact copy of *rthB* with its own promoter did not restore the production of rotihibins, which could be attributed to polar effects after the removal of the *rthB* gene and/or the insertion of the new copy of *rthB* in a region of the chromosome not optimal for its expression.

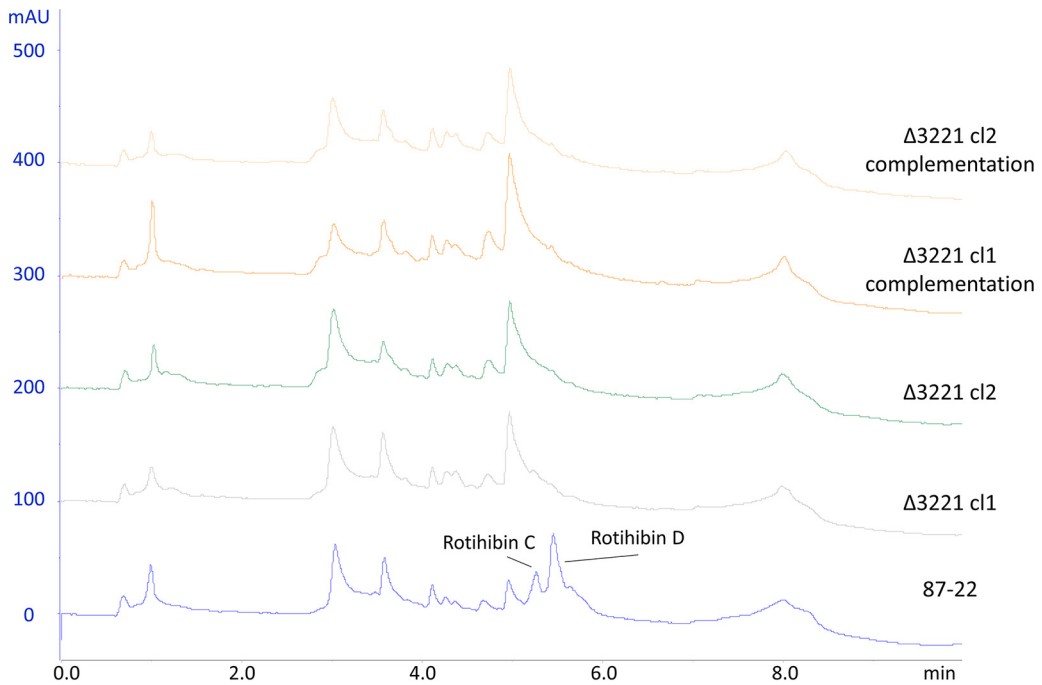

**FIG 9** Inactivation of the nonribosomal peptide gene *rthB*. Shown are data from HPLC analysis of rotihibin production by *Streptomyces scabies* 87-22 wild-type, mutant, and complementation strains. mAU, milli-absorbance unit.

**Biological activity of rotihibins C and D.** Since rotihibin A has a plant growth-inhibitory effect, we tested whether the new rotihibins C and D display similar properties. After 4 days of exposure to concentrations ranging from 0.84 to 84.4 $\mu$M rotihibin C and from 0.3 to 157.8 $\mu$M rotihibin D, the growth of duckweed (*Lemna minor* L.), especially the fronds, was suppressed compared to that in the control group (a neighboring HPLC fraction that did not display UV absorption) (Fig. 10).

Additionally, we assessed the impact of rotihibins C and D on the photochemistry of photosystem II using the $F_V/F_M$ ratio as a proxy. In dark-adapted $F_V/F_M$ measurements, the minimal fluorescence ($F_0$) is measured. Next, an intense light flash is used to close (reduce) all reaction centers and measure the maximum fluorescence, called $F_M$. The difference of $F_M - F_0$ is the $F_V$ value. The $F_V/F_M$ ratio represents the maximum potential quantum efficiency of photosystem II. In general, the greater the plant stress, the fewer open reaction centers available, which results in a lowered $F_V/F_M$ ratio. The RGB and chlorophyll fluorescence images are depicted in Fig. S2 to S5 in the supplemental material. Using the $F_V/F_M$ ratio as a proxy, the maximal photochemistry efficiency of photosystem II showed a decreasing trend with increasing rotihibin concentrations in the nutrient medium, revealing that rotihibins affect the photosynthesis of *L. minor* (Fig. 11 and 12).

The increased surface area of *L. minor* treated with 0.17 $\mu$M rotihibin C compared to the HPLC solvent indicates a hormetic effect. Hormesis is a dose-response phenomenon where a low dose of a toxic component promotes plant growth, while at higher doses, inhibition of growth is observed.

Ordinal analysis 96 h after application also showed an effect on the growth and maximal photochemistry efficiency of photosystem II ($F_V/F_M$) (52) of *Arabidopsis thaliana* plants treated with 0.1 mM rotihibins C and D (Fig. 13). Here, rotihibins were not added to the nutrient medium but were sprayed onto the plants; this explains the high concentration needed compared to the *L. minor* bioassay. Most herbicides are applied as water-based sprays, the easiest way to protect plant products for farmers. The positive control was a commercial glyphosate formulation (RoundUp Turbo) at a molar concentration of approximately 3 M. At lower concentrations of rotihibin D (0.05 mM and 0.005 mM), we observed growth promotion on *Arabidopsis* seedlings, which again reflects a hormetic effect (Fig. 14).

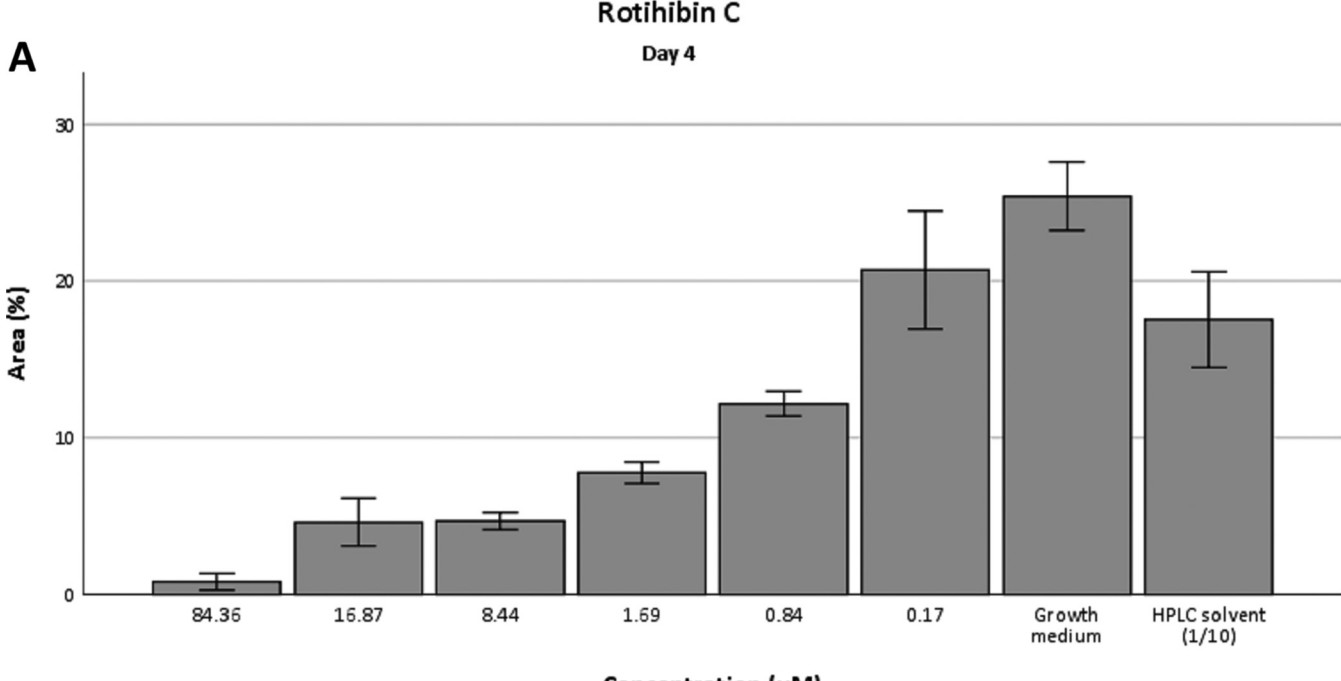

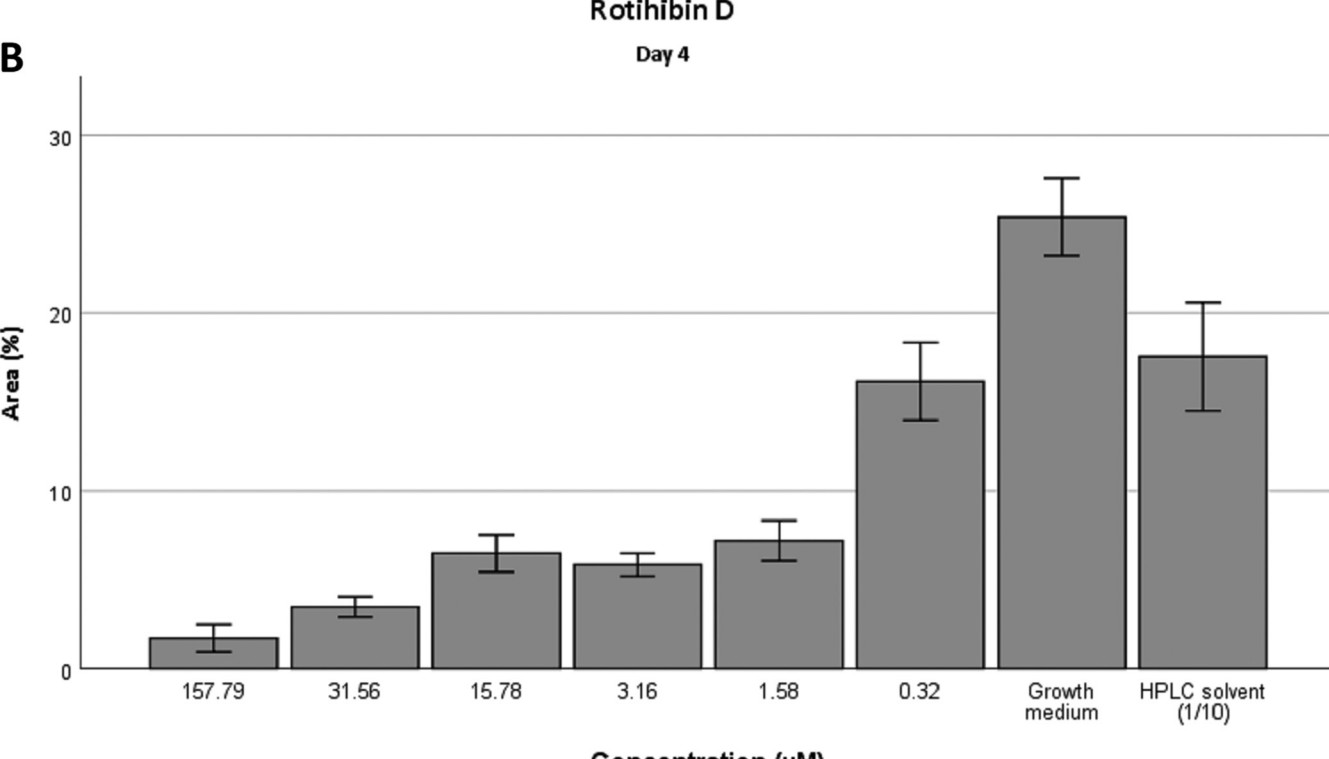

**FIG 10** Surface area measurements of *Lemna minor* treated with rotihibins. The plant growth-inhibitory activity of rotihibin C (A) and rotihibin D (B) was analyzed after 4 days compared to the control. Data are given as means from four replicates ± standard deviations (SD).

Recently, Halder et al. chemically synthesized rotihibin A and a number of variants (26). They provide evidence that rotihibin A targets the TOR (target of rapamycin) kinase (TORK) pathway, which explains the observed effects on plants. This highly conserved pathway is involved in the regulation of shoot and root development (53).

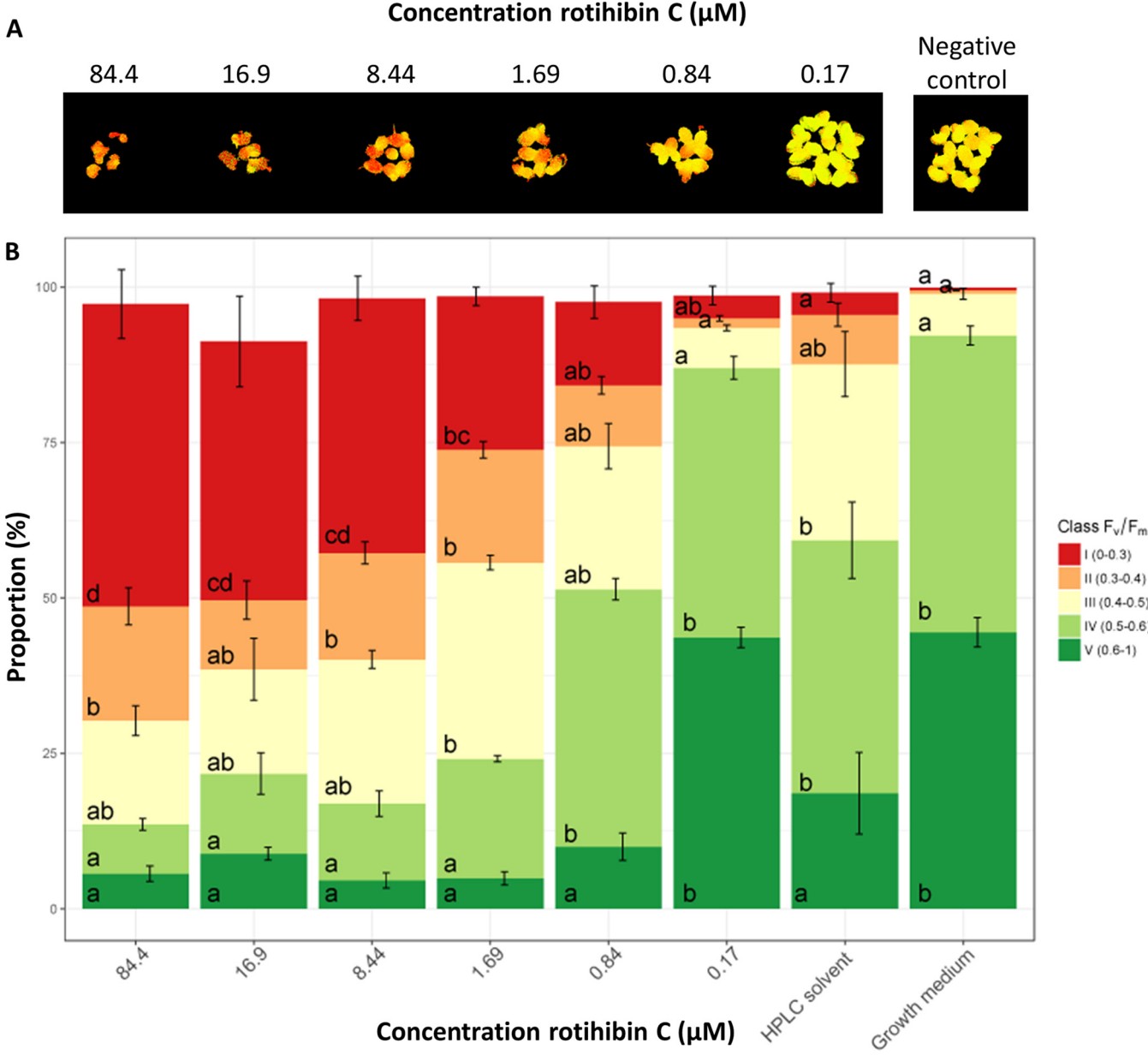

**FIG 11** Effect of rotihibin C on the maximal photochemistry efficiency of photosystem II of *Lemna minor*. (A) Chlorophyll fluorescence of *Lemna minor* fronds. Data from one out of four replicates are shown. (B) The $F_V/F_M$ values showed a decreasing trend with increasing rotihibin C concentrations. Data processing and statistical analyses were performed using Welch's *t* test and Tukey's *post hoc* test.

**Phylogenetic and evolutionary analyses of rotihibin biosynthesis.** The genomic region surrounding *rthA* was compared to all publicly available genomes using PATRIC, resulting in 24 *Actinobacteria* strains harboring a complete putative rotihibin BGC (Table 3). Rotihibin BGCs that split across more than one contig were not considered.

Strains of *S. scabies*, *S. stelliscabiei*, *S. acidiscabiei*, *S. europaeiscabiei*, and *S. ipomoeae*, all known plant pathogens, are found to contain this rotihibin biosynthetic gene cluster and are thus predicted to produce rotihibin analogues. Other rotihibin BGC-containing species, in contrast, are known as plant-protecting bacteria: *S. galbus* has been shown to induce disease resistance by the accumulation of a phytoalexin in *A. thaliana*, thereby protecting it from the anthracnose disease caused by *Colletotrichum higginsianum* (54). *S. caeruleatus* is also a biocontrol agent by inhibiting the soybean pathogen *Xanthomonas campestris* pv. glycines (55). *S. geranii* sp. nov. was isolated from the root of *Geranium carolinianum* and shares the highest 16S rRNA sequence similarity to the plant pathogen *S. turgidiscabies*

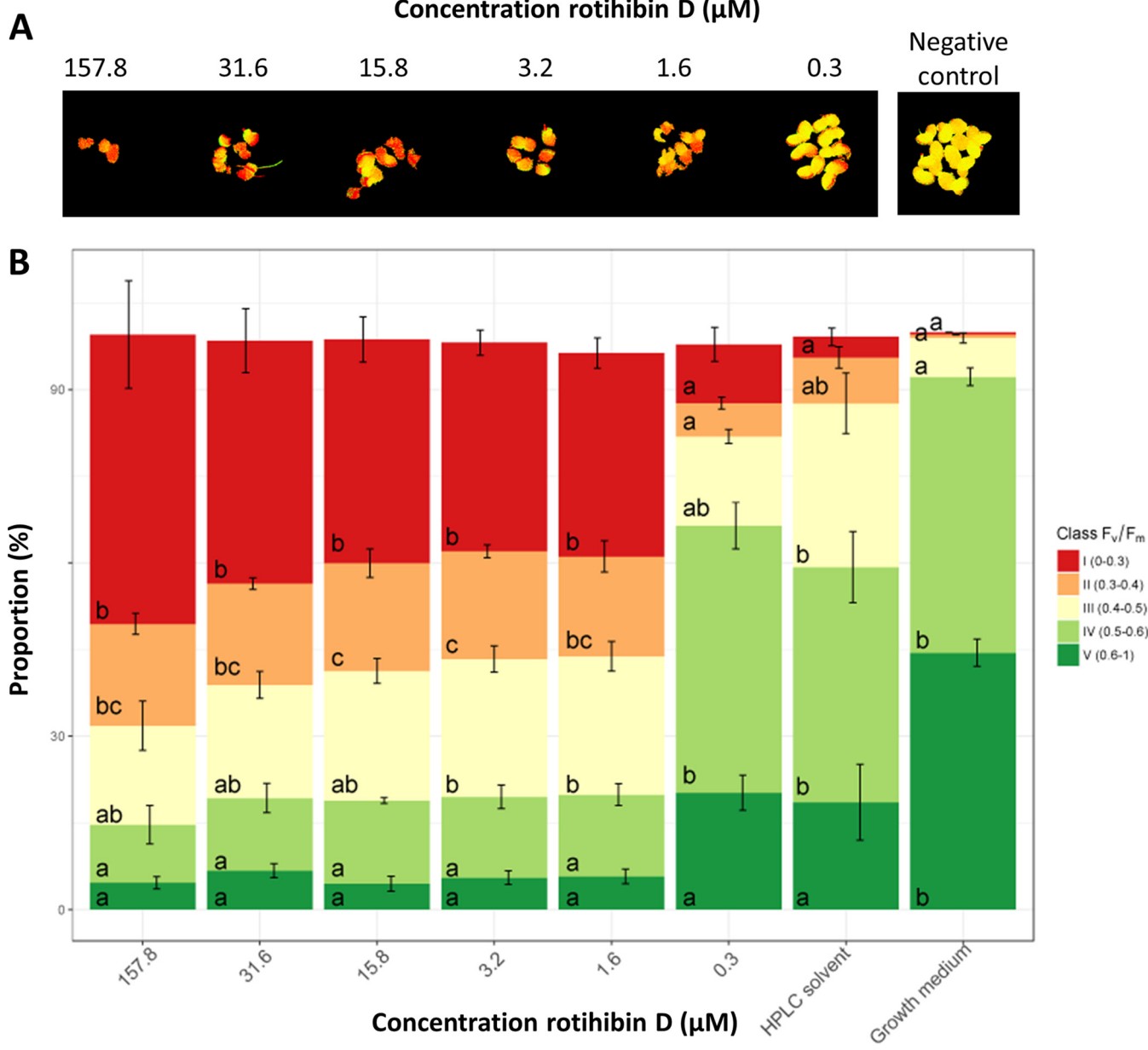

**FIG 12** Effect of rotihibin D on the maximal photochemistry efficiency of photosystem II of *Lemna minor*. (A) Chlorophyll fluorescence of *Lemna minor* fronds. Data from one out of four replicates are shown. (B) The $F_V/F_M$ values showed a decreasing trend with increasing rotihibin D concentrations. Data processing and statistical analyses were performed using Welch's $t$ test and Tukey's *post hoc* test (B).

ATCC 700248 (56). *S. hygroscopicus* subsp. *jinggangensis* strains are known to produce the antibiotic validamycin (57), while *S. corchorusii* produces butalactin (58). The latter shows biocontrol and plant growth-promoting activities and potential as a biofertilizer agent for rice plants (59). The actinomycete *Lechevalieria aerocolonigenes* has been isolated from soil in Japan and is known to produce rebeccamycin with antitumor properties (60). Curiously, the cluster is found only in plant-associated species but is not restricted to plant pathogens. Rotihibin C and D production was experimentally verified in *Streptomyces stelliscabiei* NCPPB 4040 (data not shown).

Rotihibin BGCs exist in three different organizations (Table 3), and the gene clusters identified here were classified as E-form, O-form, and X-form clusters, based on the presence of two genes: *rthE*, encoding, a luciferase-like monogygenase (LLM) class $F_{420}$-dependent oxidoreductase with unknown function, and *rthO*, which encodes a cytochrome P450 hydroxylase (Fig. 15). The cytochrome P450 OxyD from the

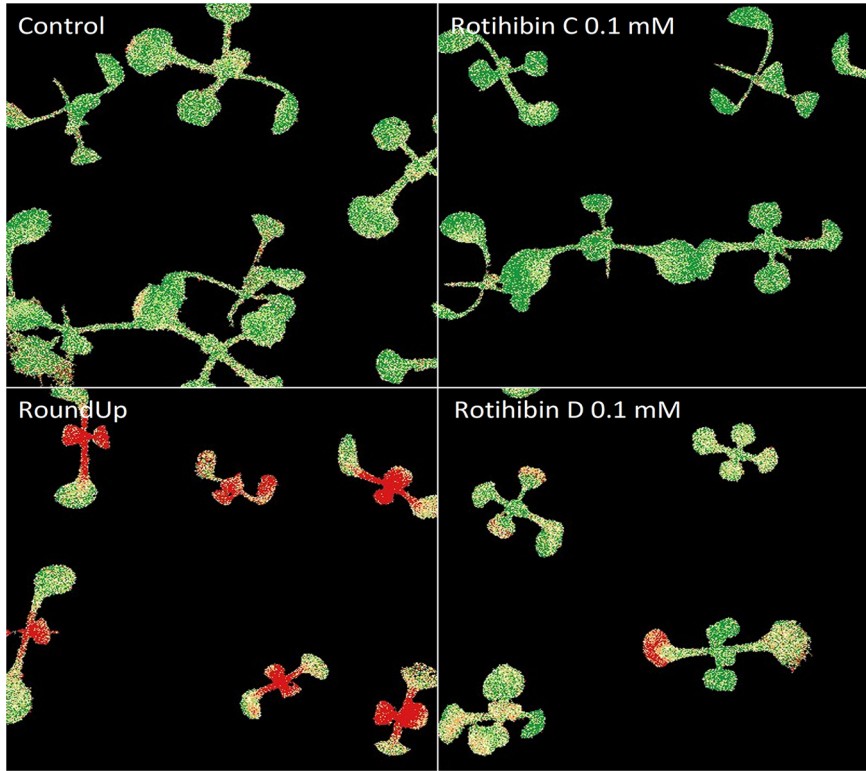

**FIG 13** Effect of rotihibins C and D on *Arabidopsis thaliana*. The spraying of plants with 0.1 mM rotihibins C and D induced a negative effect on the growth and available reaction centers of photosystem II of *Arabidopsis thaliana*.

vancomycin biosynthetic operon is involved in the biosynthesis of the modified amino acid $\beta$-hydroxytyrosine, while the cytochrome P450 TxtC was identified to be required for postcyclization hydroxylation of the cyclic dipeptide thaxtomin A (61, 62). The combination of this additional enzyme with additional modules in the *rthA* gene, which is discussed later, in *Actinobacteria* harboring the O-form BGC indicates the putative production of other rotihibin variants. The phylogenetic tree of the complete genomes correlates with the distribution of the different BGC forms (Fig. 16). The nucleotide sequences of the *rth* genes of the different *Actinobacteria* were compared with those of the *rth* genes of *Streptomyces scabies* 87-22 (Table 4). The presence of two rotihibin BGCs in *Streptomyces rameus* BK387 is quite remarkable and could be the result of recent intraspecies horizontal gene transfer (HGT) or double interspecies HGT. The loss of *rthN* in one of these BGCs could be compensated for by the other *rthN* gene.

The comparison of the *rthA* genes in the different actinomycetes displayed some NRPSs with more modules and other A-domain specificities, suggesting the production of rotihibin analogues (Table 5). These compounds are interesting and could exhibit different or better activities toward plants.

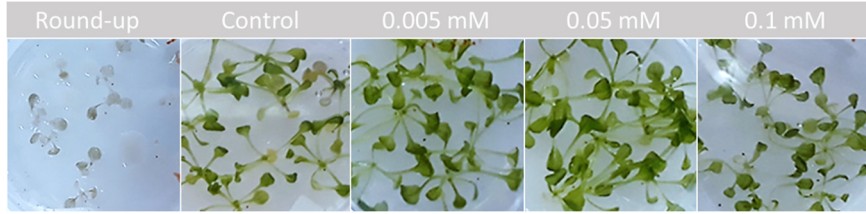

**FIG 14** Hormetic effect of rotihibin D on *Arabidopsis thaliana*. At lower concentrations (0.05 mM and 0.005 mM), there is a beneficial effect of rotihibin D on plant growth, while at a higher concentration (0.1 mM), there is a toxic effect.

**TABLE 3** Actinobacteria harboring a putative rotihibin biosynthetic gene cluster[a]

| Organism | Source | Genbank genome assembly accession no. | Presence of rotihibin BGC form | | |
|---|---|---|---|---|---|
| | | | E | O | X |
| *Streptomyces scabiei* 87-22 | | GCA_000091305.1 | ■ | | |
| *Streptomyces scabiei* NCPPB 4066 | *Solanum tuberosum*, New York City, USA | GCA_000738715.1 | ■ | | |
| *Streptomyces bottropensis* FxanaA7 | | GCA_000958545.1 | ■ | | |
| *Streptomyces bottropensis* cf124 | | GCA_900114955.1 | ■ | | |
| *Streptomyces galbus* KCCM 41354 | Tunisia | GCA_000772895.1 | ■ | | |
| *Streptomyces stelliscabiei* P3825 | | GCF_001189035.1 | ■ | | |
| *Streptomyces stelliscabiei* 1222.2 | | GCA_900215595.1 | ■ | | |
| *Streptomyces ipomoeae* B12321 | Sweet potato storage root, Louisiana, USA | GCA_006547165.1 | ■ | | |
| *Streptomyces caeruleatus* NRRL B-24802 | Tomato rhizosphere, Guangzhou, China | GCA_001514235.1 | ■ | | |
| *Streptomyces ipomoeae* 78-51 | Bunkie, Louisiana, USA | GCA_006547175.1 | ■ | | |
| *Streptomyces cinereoruber* subsp. *fructofermentans* GY16 | *Broussonetia papyrifera*, Hunan, Changsha city, China | GCA_009184865.1 | ■ | | |
| *Streptomyces europaeiscabiei* NCPPB 4086 | *Solanum tuberosum*, Ontario, Guelph, Canada | GCA_000738695.1 | ■ | | |
| *Streptomyces geranii* A301 | *Geranium carolinianum*, Emei mount, China | GCA_002954775.1 | ■ | | |
| *Streptomyces diastatochromogenes* CB02959 | Putuo Moutain, Zhejian Province, China | GCA_002803155.1 | | | ■ |
| *Streptomyces acidiscabies* NCPPB 4445 | *Solanum tuberosum* | GCA_001189015.1 | ■ | | |
| *Streptomyces ossamyceticus* JV178 | Connecticut, USA | GCA_002761895.1 | | | |
| *Streptomyces rameus* BK387, BGC 1 | | GCA_004342785.1 | | ■ | |
| *Streptomyces rameus* BK387, BGC 2 | | GCA_004342785.1 | | ■ | |
| *Lechevalieria aerocolonigenes* NRRLB-16140 | | GCA_000955955.1 | | | ■ |
| *Streptomyces hygroscopicus* subsp. *jinggangensis* 5008 | | GCA_000245355.1 | | ■ | |
| *Streptomyces hygroscopicus* subsp. *jinggangensis* TL01 | | GCA_000340845.1 | | ■ | |
| *Streptomyces corchorusii* DSM 40340 | Bangladesh | GCA_001514055.1 | | | |
| *Streptomyces jiujiangensis* NRRLS-31 | El Salvador | GCA_000718775.1 | | | |
| *Streptomyces violaceorubidus* NRRL B-16381 | | GCA_000717995.1 | | | ■ |

[a]The *rth* gene clusters were classified regarding the presence of *rthE* (E form), *rthO* (O form), or neither (X form).

## MATERIALS AND METHODS

**Fermentation and analysis of *Streptomyces scabies* RL-34, 87-22, and mutant strains.** *Streptomyces scabies* RL-34, *S. scabies* 87-22, and the Δ*rthB* mutant were cultured in a shaking incubator at 250 rpm at 28°C for 4 days using borosilicate-baffled flasks with membrane screw caps containing 100 ml of ISP-4 medium, prepared as follows: 10 g soluble starch (Sigma-Aldrich), 1 g $K_2HPO_4$, 1 g $MgSO_4$, 1 g NaCl, 2 g $(NH_4)_2SO_4$, 2 g $CaCO_3$, 1 mg $FeSO_4$, 1 mg $MnCl_2$, and 1 mg $ZnSO_4$ were dissolved in 1 liter of deionized water. For the induction of rotihibin production, wild-type cultures were grown in the presence of 0.7% cellobiose (Sigma-Aldrich) or 0.5% glucose (Merck).

Cultures were centrifuged at 10,000 × *g* for 10 min, and 2 ml of each supernatant was transferred to a new tube. The samples were mixed with *n*-butanol (1:1), vortexed gently for 10 min, and centrifuged for 15 min at 16,000 × *g*. The aqueous phase of the extract was dried and stored at 20°C until further use.

The dried extract was dissolved in 10 mM ammonium formate ($NH_4FA$) (pH 3) and analyzed by HPLC, monitoring the absorbance at 220 nm. The sample was eluted with solvent A (10 mM $NH_4FA$, pH 3) and solvent B (acetonitrile [ACN]) using a 2-to-90% 7.5-min gradient at 1 ml/min on a Zorbax Eclipse Plus $C_{18}$ column (4.6 by 100 mm, 3.5 $\mu$m). Fractions were collected every 15 s, corresponding to 250 $\mu$l. The total area under the curve (AUC) for each metabolite was calculated. Student's *t* test (two tailed, homoscedastic) was executed to evaluate the differential metabolite abundances between the different conditions in a statistical way.

The fractions of interest were loaded into a Triverse Nanomate source (Advion) and, using 1.6 kV and 0.3 lb/in$^2$ of pressure, electrosprayed into a Waters Synapt G1 mass spectrometer. Scans typically ranged from *m/z* 100 to 1,200, with a scan time of 1 s, for a total of 2 min. For MS/MS experiments, the collision energy was optimized to achieve good fragmentation spectra.

**Generation of the Δ*rthB* (*scab_3221*; *scab_RS01465*) deletion mutant in *Streptomyces scabies* 87-22.** Deletion of the *rthB* gene was performed in *S. scabies* 87-22 using the ReDirect PCR targeting method (63) by replacing the genomic copy of the gene with an apramycin resistance cassette: using a

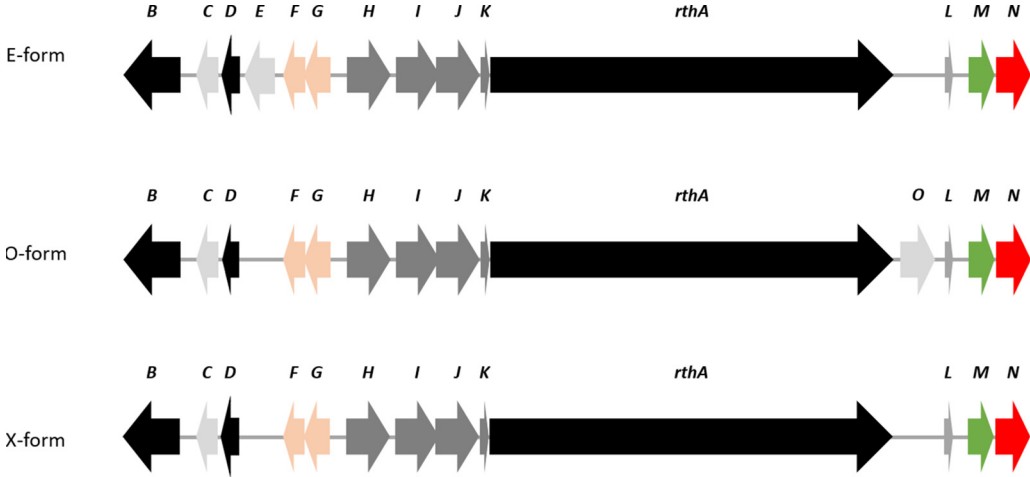

**FIG 15** Schematic representation of E-, O-, and X-form rotihibin biosynthetic gene clusters.

HindIII-EcoRI restriction fragment obtained from pIJ773 (Table 6) as the PCR template, a disruption cassette, containing the resistance gene *aac(3)IV* and *oriT* for conjugative transfer, was amplified by PCR with primers BDF33 and BDF34 and gel purified as performed previously for *cebR*, *cebE*, *msiK*, and *bglC* (11, 13, 29).

In parallel, cosmid 2012 (cos2012), based on Supercos-1 and containing a genomic insert of *S. scabies* 87-22 ranging from positions 341913 to 386384, including *rthB*, was introduced by electroporation into *Escherichia coli* BW25113 carrying plasmid pIJ790 for the arabinose-inducible expression of the λ Red recombination machinery. After culture to an optical density (OD) of 0.4 with arabinose at 10 mM to induce the λ Red machinery, 50 μl of washed *E. coli* BW25113/pIJ790 cos2012 cells was electroporated with 100 ng (in 1 μl) of the *rthB*-targeting disruption cassette, and apramycin-resistant clones were selected on LB agar plates plus apramycin (50 μg/ml) and kanamycin (50 μg/ml). The gene replacement by the disruption cassette on the cosmid was confirmed by PCR using primers BDF35 and BDF36 (Table 6) and Sanger sequencing. The resulting cos2012Δ3221 construct was purified using a GeneJET plasmid miniprep kit (Thermo Scientific) (according to the manufacturer's guidelines) and transferred by intergeneric conjugation with *E. coli* ET12567/pUZ8002 (ETpUZ) as the donor strain and *S. scabies* 87-22 as the recipient strain. After growing ETpUZ to an OD of 0.4 (50-ml culture in LB plus chloramphenicol [30 μg/ml], apramycin [50 μg/ml], kanamycin [50 μg/ml], and ampicillin [100 μg/ml]), cells were washed to remove antibiotics and mixed with *S. scabies* 87-22 spores for mating. The conjugative mixture was then plated on 30-ml petri dishes of soy flour mannitol medium (SFM) (20 g/liter mannitol, 20 g/liter soy flour, and 20 g/liter agar in tap water and autoclaved twice) plus 10 mM MgCl₂ and overlaid with nalidixic acid (50 μg/ml) and apramycin (40 μg/ml). Exconjugants were transferred to ISP-4 agar plates plus nalidixic acid (25 μg/ml) and apramycin (50 μg/ml) to obtain uniform mutant lines that were then used to prepare spore stocks. Each *S. scabies* ΔrthB mutant was checked for apramycin resistance and kanamycin sensitivity on ISP-2 agar plates. Genomic DNA was extracted from 48-h liquid cultures in tryptic soy broth (TSB) using a GenElute bacterial genomic DNA kit (Sigma-Aldrich) (according to the manufacturer's guidelines) and used as the PCR template for mutation confirmation.

**Complementation of the ΔrthB deletion mutant.** *rthB*, including its upstream (−701 bp) and downstream (+501 bp) regions, was amplified by PCR with primers BDF35 and BDF36 to obtain a 3,512-bp fragment flanked by two XbaI restriction sites. The PCR product was first cloned into a pJET1.2 vector, which was named pBDF054. Using the XbaI enzyme, *rthB* and surrounding regions were isolated and cloned into a pAU3-45 (64) integrative vector named pBDF043. Conjugation was performed with *S. scabies* ΔrthB clones 1 and 2 on SFM (plus 10 mM MgCl₂) overlaid with apramycin (50 μg/ml), nalidixic acid (50 μg/ml), and thiostrepton (12.5 μg/ml). The integration of the plasmid was confirmed by PCR with primers BDF69 and BDF70.

**Structural confirmation by NMR spectroscopy.** Both rotihibin analogues were dissolved in 0.6 ml of DMSO-d₆, and the nuclear magnetic resonance (NMR) spectra were recorded at 25°C on a Bruker Avance II 700-MHz spectrometer equipped with a 5-mm Prodigy TCI N₂ cooled cryoprobe. Total correlated spectroscopy (TOCSY) spectra were recorded using the standard pulse sequences from the Bruker library.

**Protein extraction and sample preparation.** *S. scabies* RL-34 cells were cultivated in three different growth media (ISP-4 medium, ISP-4 medium supplemented with 0.7% cellobiose, and ISP-4 medium supplemented with 0.5% glucose) for 96 h (28°C at 250 rpm), with three biological replicates under each condition. The cultures were centrifuged at 10,000 × *g* for 10 min. The resulting pellet was washed two times and resuspended in 10 ml lysis buffer (1× phosphate-buffered saline [PBS], 0.1% SDS, protease inhibitor cocktail). The homogeneous mycelium suspension was sonicated (20 times with 30 s on and 30 s off at 30%) on ice using a Branson digital sonifier 450 cell disruptor. The homogenate was centrifuged at

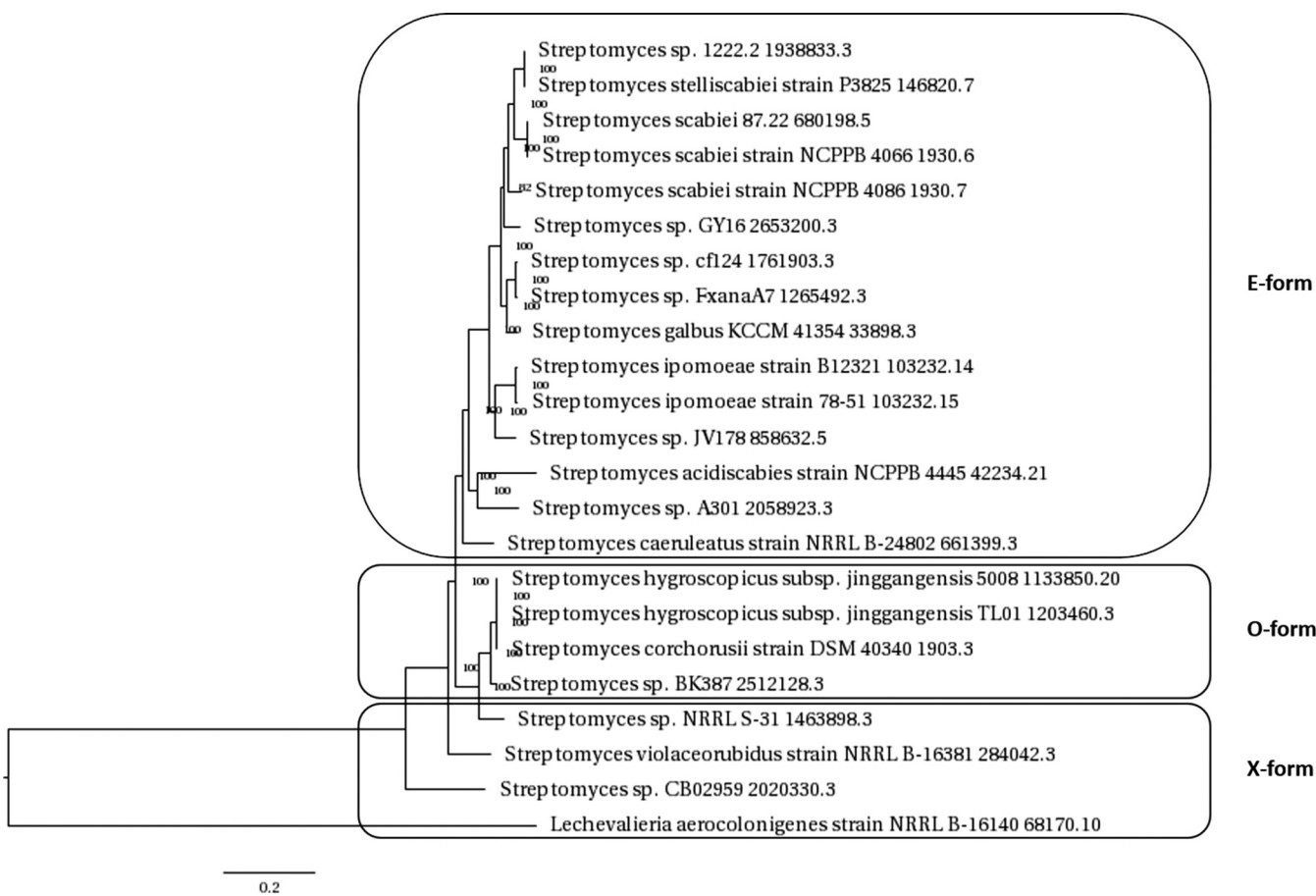

**FIG 16** Phylogenetic tree of 23 *Actinobacteria* analyzed in this study. The phylogenetic tree was generated using the codon tree method in PATRIC.

16,000 × *g* for 30 min at 4°C. Pellets were discarded, and the supernatant was subjected to trichloroacetic acid (TCA) precipitation. The crude intracellular extract was mixed with a 100% (wt/vol) TCA solution (Sigma-Aldrich) (4:1) and incubated overnight. This mixture was centrifuged at 16,000 × *g* for 15 min at 4°C, and the protein pellet was washed two times with ice-cold acetone and solubilized in 2 M urea in 50 mM ammonium bicarbonate. The protein concentration was determined using the Pierce Coomassie protein assay kit.

Protein extracts (10 μg) were spiked with bovine serum albumin (BSA) (MS-grade protein standard) (0.2 ng/μl) as an internal standard. The proteins were subsequently reduced with 2.5 mM dithiothreitol (DTT) (10 min at 60°C), alkylated with 7.5 mM iodoacetamide (20 min at room temperature), and trypsinized (1/50) (overnight at 37°C). The reaction was stopped with 1% HCOOH. After centrifugation (30 min at 13,200 rpm), the samples were filtered with a prewashed 0.22-μm Costar filter and transferred to clean tubes.

**Targeted proteomic analysis.** The resulting peptide solutions were injected (0.1 μg) onto an ultra-performance liquid chromatography (UPLC) M-Class system where the peptides were trapped for 5 min, at 15 μl/min, on a 300-μm by 50-mm, 5-μm, 100-Å Acquity UPLC M-Class symmetry $C_{18}$ trap column (Waters) and separated on the iKey separation device (150 μm by 100 mm, 1.8 μm) (HSS T3; Waters) with solvent A (0.1% HCOOH in $H_2O$ [Biosolve]) and solvent B (0.1% HCOOH in ACN [Biosolve]) using a 20-min gradient of 3 to 50% solvent B at a flow rate of 1 μl/min. The strong and weak solutions used to wash the autosampler were 0.1% HCOOH in $H_2O$ and 0.1% HCOOH in acetonitrile-water-isopropanol (50:25:25, vol/vol/vol), respectively. The separated peptides were introduced into the IonKey source coupled to a Waters Xevo TQ-S triple-quadrupole mass spectrometer for quantification of the analytes in the positive-ion mode (ESI$^+$). The ESI-MS/MS parameters were as follows: capillary voltage of 3.6 kV, cone voltage of 40 V, source temperature of 120°C, and collision gas argon flow rate of 0.19 ml/min. The transitions of the selected precursor ions were detected in MRM mode at different collision energies specific for each precursor (Table 7). Data were acquired using MassLynx 4.1 software and processed in Skyline 3.7. The transition curves were subjected to Savitsky-Golay smoothing, and their area under the curve (AUC) was determined and normalized to the spiked BSA standard. Student's *t* test (two tailed, homoscedastic) was executed to evaluate the differential protein abundances between the different conditions in a statistical way.

**Bioinformatics analysis of the rotihibin biosynthetic gene cluster.** Prediction of the gene cluster of *Streptomyces scabies* 87-22 was performed using antiSMASH 3.0 (30). The sequence of the

**TABLE 4** Nucleotide sequence comparison of *rth* genes[a]

| Organism | % identity | | | | | | | | | | | | | | |
|---|---|---|---|---|---|---|---|---|---|---|---|---|---|---|---|
| | rthB | rthC | rthD | rthE | rthF | rthG | rthH | rthI | rthJ | rthK | rthA | rthO | rthL | rthM | rthN |
| *Streptomyces scabiei* NCPPB 4066 | 98.63 | 99.64 | 99.16 | 99.05 | 99.75 | 99.33 | 99.80 | 98.23 | 99.53 | 99.56 | 99.64 | X | 99.52 | 99.40 | 98.98 |
| *Streptomyces galbus* KCCM 41354 | 81.96 | 92.73 | 91.60 | 91.47 | 92.72 | 90.48 | 89.89 | 87.91 | 89.91 | 91.98 | 89.49 | X | 93.33 | 93.81 | 90.78 |
| *Streptomyces bottropensis* cf124 | 80.51 | 93.33 | 92.58 | 91.47 | 91.79 | 90.61 | 88.71 | 85.40 | 90.13 | 92.71 | 89.08 | X | 93.81 | 93.51 | 90.31 |
| *Streptomyces bottropensis* FxanaA7 | 80.47 | 93.33 | 92.58 | 91.29 | 91.79 | 90.75 | 88.78 | 85.34 | 90.19 | 92.71 | 88.74 | X | 93.33 | 93.61 | 90.31 |
| *Streptomyces ipomoeae* B12321 | 90.65 | 93.45 | 89.08 | 87.73 | 86.83 | 86.86 | 85.60 | 82.34 | 85.39 | 89.82 | 86.69 | X | 93.81 | 94.42 | 92.51 |
| *Streptomyces cinereoruber* subsp. *fructofermentans* GY16 | 81.70 | 92.24 | 88.64 | 89.02 | 89.65 | 85.77 | 88.91 | 84.98 | 87.39 | 91.58 | 86.68 | X | 94.76 | 93.00 | 89.76 |
| *Streptomyces ipomoeae* 78-51 | 88.48 | 93.45 | 88.80 | 87.81 | 86.96 | 86.45 | 85.54 | 82.10 | 85.10 | 89.92 | 86.69 | X | 93.81 | 94.62 | 92.36 |
| *Streptomyces caeruleatus* NRRL B-24802 | 87.05 | 94.06 | 89.08 | 89.57 | 86.38 | 86.86 | 84.99 | 83.23 | 85.18 | 90.53 | 87.35 | X | 94.76 | 93.61 | 88.55 |
| *Streptomyces europaeiscabiei* NCPPB 4086 | 82.14 | 90.42 | 84.17 | 88.20 | 89.64 | 89.16 | 90.15 | 83.78 | 85.78 | 90.00 | 87.31 | X | 93.81 | 94.21 | 92.04 |
| *Streptomyces stelliscabiei* 1222.2 | 80.84 | 92.12 | 86.33 | 91.05 | 89.52 | 89.03 | 87.34 | 86.04 | 86.92 | 90.53 | 87.31 | X | 91.22 | 85.68 | 89.20 |
| *Streptomyces stelliscabiei* P3825 | 81.11 | 92.12 | 86.33 | 89.48 | 89.52 | 89.03 | 87.34 | 86.12 | 86.75 | 90.35 | 88.67 | X | 91.22 | 85.58 | 89.05 |
| *Streptomyces geranii* A301 | 81.26 | 90.67 | 88.52 | 88.09 | 87.25 | 86.57 | 84.48 | 84.68 | 85.14 | 93.09 | 85.30 | X | 92.31 | 88.14 | 90.78 |
| *Streptomyces diastatochromogenes* CB02959 | 84.58 | 92.12 | 87.54 | X | 87.69 | 86.80 | 84.77 | 84.68 | 86.66 | 85.82 | 86.21 | X | 73.27 | 87.61 | 82.58 |
| *Streptomyces ossamyceticus* JV178 | 77.56 | 87.39 | 83.29 | 84.02 | 87.25 | 81.77 | 74.33 | 79.70 | 82.18 | 84.21 | 83.37 | X | 90.95 | 81.40 | 78.38 |
| *Streptomyces acidiscabies* NCPPB 4445 | 73.55 | 85.21 | 80.12 | 81.44 | 81.07 | 79.50 | 73.78 | 76.22 | 78.68 | 88.11 | 80.29 | X | 74.63 | 91.71 | 86.77 |
| *Streptomyces rameus* BK387 | 74.27 | 82.67 | 74.63 | X | 73.81 | 75.83 | 74.42 | 74.79 | 75.19 | 81.46 | 77.98 | ✓ | 81.95 | 76.86 | 75.06 |
| *Streptomyces hygroscopicus* subsp. *jinggangensis* TL01 | 74.69 | 81.14 | 73.56 | X | 74.58 | 75.79 | 73.76 | 74.41 | 74.29 | 81.44 | 77.67 | ✓ | 81.46 | 76.37 | 74.55 |
| *Streptomyces jiujiangensis* NRRL S-31 | 74.51 | 82.47 | 72.40 | X | 73.71 | 71.37 | 73.88 | 75.35 | 75.41 | 78.83 | 77.72 | ✓ | 81.60 | 77.29 | 76.83 |
| *Streptomyces hygroscopicus* subsp. *jinggangensis* 5008 | 74.69 | 81.14 | 73.56 | X | 74.58 | 75.79 | 73.76 | 74.41 | 74.29 | 80.98 | 75.48 | ✓ | 81.46 | 76.37 | 74.55 |
| *Streptomyces corchorusii* DSM 40340 | 74.91 | 81.14 | 73.82 | X | 74.58 | 75.79 | 73.82 | 74.32 | 74.45 | 80.49 | 77.64 | ✓ | 73.54 | 76.37 | 74.71 |
| *Lechevalieria aerocolonigenes* NRRL B-16140 | 76.87 | 77.60 | 75.60 | X | 72.68 | 70.83 | 70.05 | 75.35 | 73.86 | 78.46 | 71.03 | X | 83.25 | 73.02 | 74.00 |
| *Streptomyces violaceorubidus* NRRL B-16381 | 72.89 | 80.73 | 73.54 | X | 71.47 | 76.28 | 70.05 | 69.96 | 68.04 | 72.83 | 74.81 | X | 72.38 | 73.48 | 75.66 |
| *Streptomyces rameus* BK387 | 71.70 | 79.76 | 72.63 | X | 73.37 | 68.18 | 71.18 | 73.02 | 74.02 | 73.08 | 73.39 | ✓ | 73.54 | 74.03 | X |

[a]The nucleotide sequences of *rth* genes of different *Actinomycetes* were compared against those of *Streptomyces scabies* 87-22 using BLAST. X: gene is missing, checkmark: gene is present.

**TABLE 5** Comparison of *rthA* genes in different *Actinomycetes*[a]

| Strain | No. of modules | Length (aa) | Analogue | | | | | | |
|---|---|---|---|---|---|---|---|---|---|
| | | | Module 1 | Module 2 | Module 3 | Module 4 | Module 5 | Module 6 | Module 7 |
| *Streptomyces violaceorubidus* NRRL B-16381 | 4 | 4,235 | X | | NosA-M2-Ser | No hit | MycC-M2-Asx | X | X |
| *Streptomyces caeruleatus* NRRL B-24802 | 5 | 5,490 | Pps4-M2-Glu | | PvdD-M3-Ser | Cda2-M3-Asn | Cda2-M3-Asn | X | X |
| *Streptomyces cinereoruber* subsp. *fructofermentans* GY16 | 5 | 5,514 | Pps4-M2-Glu | | PvdD-M3-Ser | Cda2-M3-Asn | Cda2-M3-Asn | X | X |
| *Streptomyces geranii* A301 | 5 | 5,523 | Pps4-M2-Glu | | PvdD-M3-Ser | Cda2-M3-Asn | Cda2-M3-Asn | X | X |
| *Streptomyces ipomoeae* 78-51 | 5 | 5,458 | Pps4-M2-Glu | | PvdD-M3-Ser | Cda2-M3-Asn | Cda2-M3-Asn | X | X |
| *Streptomyces ipomoeae* 88-35 | 5 | 5,458 | Pps4-M2-Glu | | PvdD-M3-Ser | Cda2-M3-Asn | Cda2-M3-Asn | X | X |
| *Streptomyces ipomoeae* B12321 | 5 | 5,458 | Pps4-M2-Glu | | PvdD-M3-Ser | Cda2-M3-Asn | Cda2-M3-Asn | X | X |
| *Streptomyces scabies* 87-22 | 5 | 5,514 | Pps4-M2-Glu | | PvdD-M3-Ser | Cda2-M3-Asn | Cda2-M3-Asn | X | X |
| *Streptomyces scabies* NCPPB 4066 | 5 | 5,510 | Pps4-M2-Glu | | PvdD-M3-Ser | Cda2-M3-Asn | Cda2-M3-Asn | X | X |
| *Streptomyces scabies* RL-34 | 5 | 5,535 | Pps4-M2-Glu | | PvdD-M3-Ser | Cda2-M3-Asn | Cda2-M3-Asn | X | X |
| *Streptomyces bottropensis* cf124 | 5 | 5,517 | Pps4-M2-Glu | | PvdD-M3-Ser | Cda2-M3-Asn | Cda2-M3-Asn | X | X |
| *Streptomyces bottropensis* FxanaA7 | 5 | 5,517 | Pps4-M2-Glu | | PvdD-M3-Ser | Cda2-M3-Asn | Cda2-M3-Asn | X | X |
| *Streptomyces acidiscabies* NCPPB 4445 | 5 | 5,411 | PvdD-M3-Ser | | PvdD-M3-Ser | Cda2-M3-Asn | Cda2-M3-Asn | X | X |
| *Streptomyces galbus* KCCM 41354 | 5 | 5,478 | Pps4-M2-Glu | | PvdD-M3-Ser | No hit | Cda2-M3-Asn | X | X |
| *Streptomyces europaeiscabiei* NCPPB 4086 | 5 | 5,449 | Pps4-M2-Glu | | PvdD-M3-Ser | No hit | Cda2-M3-Asn | X | X |
| *Streptomyces stelliscabiei* 1222.2 | 5 | 5,507 | Pps4-M2-Glu | | Pps4-M2-Glu | Cda2-M3-Asn | Cda2-M3-Asn | X | X |
| *Streptomyces stelliscabiei* P3825 | 5 | 5,504 | Pps4-M2-Glu | | Pps4-M2-Glu | Cda2-M3-Asn | Cda2-M3-Asn | X | X |
| *Lechevalieria aerocolonigenes* NRRL B-16140 | 7 | 7,050 | Pps4-M2-Glu | | Pps4-M2-Glu | Cda2-M3-Asn | Cda2-M3-Asn | TycC-M2-Gln | Pps4-M2-Glu |
| *Streptomyces diastatochromogenes* CB02959 | 7 | 7,066 | Pps4-M2-Glu | | PvdD-M3-Ser | Cda2-M3-Asn | Cda2-M3-Asn | | Pps4-M2-Glu |
| *Streptomyces ossamyceticus* JV178 | 7 | 7,399 | Pps4-M2-Glu | | PvdD-M3-Ser | Cda2-M3-Asn | MycC-M2-Asx | No hit | Pps4-M2-Glu |
| *Streptomyces jiujiangensis* S-31 | 7 | 7,036 | No hit | | PvdD-M3-Ser | Cda2-M3-Asn | Cda2-M3-Asn | AcmB-M2-Val | Pps4-M2-Glu |
| *Streptomyces corchorusii* DSM 40340 | 7 | 7,027 | No hit | | Pps4-M2-Glu | Cda2-M3-Asn | Cda2-M3-Asn | AcmB-M2-Val | Pps4-M2-Glu |
| *Streptomyces hygroscopicus* subsp. *jinggangensis* 5008 | 7 | 7,027 | No hit | | Pps4-M2-Glu | Cda2-M3-Asn | Cda2-M3-Asn | AcmB-M2-Val | Pps4-M2-Glu |
| *Streptomyces hygroscopicus* subsp. *jinggangensis* TL01 | 7 | 7,027 | No hit | | Pps4-M2-Glu | Cda2-M3-Asn | Cda2-M3-Asn | AcmB-M2-Val | Pps4-M2-Glu |
| *Streptomyces rameus* BK387 | 7 | 7,033 | No hit | | Pps4-M2-Glu | Cda2-M3-Asn | Cda2-M3-Asn | AcmB-M2-Val | Pps4-M2-Glu |
| *Streptomyces rameus* BK387 | 7 | 7,379 | Pps4-M2-Glu | | PvdD-M3-Ser | Cda2-M3-Asn | MycC-M2-Asx | AcmB-M2-Asx | Pps4-M2-Glu |

[a]The different modular organizations and A-domain specificities of the nonribosomal peptide synthase (NRPS) RthA suggest the production of rotihibin analogues (grouped together) over the investigated strains. aa, amino acids.

**TABLE 6** Primers and genetic constructs used in this study[a]

| Primer or genetic construct | Sequence (5′→3′) or description | Application, reference, or source |
|---|---|---|
| **Primers** | | |
| BDF33 (ReDirect *SCAB_3221_f_+3*) | GGCCGTTCCGCCGCCCCGGCACTCCCACCCGGGGCGGTG<u>ATTCCGGGGATCCGTCGACC</u> | Disruption cassette amplification |
| BDF34 (ReDirect *SCAB_3221_r_+2307*) | AACACCGGCCGCCGGCCCGCCCGTACCGCCGACGCTTCA<u>TGTAGGCTGGAGCTGCTTC</u> | Disruption cassette amplification |
| BDF35 (*SCAB_3221_f_-682_XbaI*) | AAA*TCTAGA*GCACGAACACGAGAAGACC | PCR checking/ complementation |
| BDF36 (*SCAB_3221_r_+2791_XbaI*) | AAA*TCTAGA*TTGGGAGACGATCTTGATGC | PCR checking/ complementation |
| BDF69 (*SCAB_3221_fwd_+985*) | CCGCCTATGTCATCTACACC | PCR checking |
| BDF70 (*SCAB_3221_rev_+1097*) | CCGAAGCCGTACTCCTCG | PCR checking |
| **Plasmids or cosmids** | | |
| pIJ773 | Template for amplification of the apramycin resistance cassette [*aac(3)IV* (Apra[r]) *oriT* (RK2) FRT *amp* (Amp[r])] | 63 |
| cos2012 | Supercos-1 (Agilent) derivative containing the genomic insert of *S. scabies* 87-22 from positions 341913–386384 (Amp[r] Kan[r]) | Isolde Francis |
| pIJ790 | Contains the λ Red recombination under the control of an arabinose-inducible promoter (p$_{araBAD}$) (Cml[r]) | 63 |
| cos2012Δ3221 | Cosmid 2012 derivative containing the apramycin resistance cassette instead of the *rthB* gene | This study |
| pUZ8002 | Nontransmissible plasmid supplying transfer functions for mobilization of *oriT*-containing vectors from *E. coli* to *Streptomyces* spp. (Kan[r]) | 80 |
| pJET1.2/blunt | *E. coli* plasmid used for high-efficiency blunt-end cloning of PCR products (Amp[r]) | Thermo Scientific |
| pBDF054 | pJET1.2 derivative containing the 3,512-bp DNA fragment for *rthB* complementation | This study |
| pAU3-45 | pSET152 derivative, integrative plasmid with a thiostrepton resistance gene inserted into the blunted NheI restriction site [*lacZα ori* (pUC18) *aac(3)IV* (Apra[r]) *oriT* (RK2) *attP* (φC31) *int* (φC31) *tsr* (Thio[r])] | 64 |

[a]Engineered restriction sites are indicated in italic and underlined, non-homologous extensions are underlined.

biosynthetic rotihibin gene cluster was further analyzed and annotated using UniProt and BLAST (65, 66). Similar gene clusters were identified in other *Actinomycetes* strains by PATRIC 3.5.21 analysis (31). A phylogenetic tree was calculated, from the *rthA* MUSCLE sequence alignment, using the neighbor-joining method with MEGA-X (67). Thaxtomin synthetase A (*txtA*) was selected as the outgroup.

The module/domain organization of the different NRPSs in the rotihibin gene cluster was predicted via PRISM (68, 69). The A-domain specificity was predicted via four different software tools: the LSI-based A-domain function predictor, NRPSsp, the PKS/NRPS Web server, and SEQL-NRPS. The LSI-based predictor uses latent semantic indexing to predict adenylation domain specificities (70). NRPSsp uses hits against hidden Markov model (HMM) databases to predict specificities of NRPS adenylation domains (71). The PKS/NRPS Web server uses BLAST to detect catalytic domains in NRPS and predicts A-domain specificities by comparing signatures of A domains with those of known substrates (72). Finally, SEQL-NRPS predicts A-domain specificities using the discriminative classification method sequence learner (SEQL) (73).

***Lemna minor* L. and *Arabidopsis thaliana* L. Heynh. bioassay.** A sterile 24-well plate was used to grow *Lemna minor* (duckweed) in 2 ml mineral medium [11.1 mg CaCl$_2$, 202 mg KNO$_3$, 49.6 mg MgSO$_4$·7H$_2$O, 50.3 mg KH$_2$PO$_4$, 27.8 mg K$_2$HPO$_4$, 6 mg FeSO$_4$·7H$_2$O, 17.4 mg K$_2$SO$_4$, 5.72 mg H$_3$BO$_3$, 2.82 mg MnCl$_2$·4H$_2$O, 0.6 mg ZnSO$_4$, 10 mg Na$_2$-EDTA, 0.008 mg CuCl$_2$·H$_2$O, 0.054 mg CoCl$_2$·6H$_2$O, and 0.043 mg (NH$_4$)$_6$Mo$_7$O$_{24}$ dissolved in 1 liter of MilliQ (MQ) water (pH 6.5 ± 0.1)] (74). The plants were incubated in quadruple for 4 days in a growth chamber (16-h light exposure at 22°C). In order to assess the sensitivity of *L. minor* to rotihibins C and D, a wide concentration range (0.17 to 84.4 μM for rotihibin C and 0.3 to 157.6 μM for rotihibin D) was tested. After 4 days of incubation, the plates were analyzed based on the growth and maximal photochemistry efficiency of photosystem II via chlorophyll fluorescence ($F_V/F_M$) of the duckweed plants.

*Arabidopsis thaliana* ecotype Col-0 pregerminated seeds were transferred to petri dishes with half-strength Murashige-Skoog (MS) medium without sucrose (0.22% MS salts and 0.8% plant agar) (75). Nine days after germination, the plants were treated with rotihibins C and D dissolved in water (0.005 to 0.1 mM) via nebulization on the leaves. After 4 days of treatment, the plates were analyzed based on the growth and $F_V/F_M$ values of the *Arabidopsis* plants.

Imaging was achieved with an in-house-developed phenotyping platform, in an environment-controlled growth chamber, property of the Ghent University Laboratory of Applied Mycology and Phenomics. The platform allows visualization of diverse physiological traits via a multispectral 3CCD camera equipped with 12 interference filters in real time, based on specific absorption, reflection, and emission patterns, such as leaf surface, efficiency of photosynthesis, chlorophyll and anthocyanin contents, and green fluorescent protein (GFP)-tagged organisms. This platform is equipped with a dispenser, which can be fitted with a nozzle to treat the plants with rotihibins or other agrochemicals in a standardized manner. Image data processing was performed using Data Analysis (version 5.4.6; Phenovation, Wageningen, Netherlands), and statistical analysis was performed in RStudio (version

**TABLE 7** Transitions selected for MRM validation of Rth proteins upon cellobiose and glucose supply[a]

| Peptide | Precursor ion mass (m/z) | CE (V) | Fragment ion mass (m/z) |
|---|---|---|---|
| **RthA** | | | |
| AWIDSDLATPVPVTGER | 913.968++ | 33 | L [y11]—1,139.642+ |
| | | | A [y10]—1,026.558+ |
| | | | T [y9]—955.521+ |
| ADTSGDPTFEELLDR | 833.384++ | 30 | P [y9]—1,119.568+ |
| | | | T [y8]—1,022.515+ |
| | | | F [y7]—921.468+ |
| GGTVPFAVPAALR | 628.362++ | 22 | A [y9]—941.557+ |
| | | | V [y8]—844.504+ |
| | | | P [y7]—697.436+ |
| **RthH** | | | |
| VTDEQLAALDLSR | 715.878++ | 25 | Q [y9]—986.563+ |
| | | | L [y8]—858.504+ |
| | | | A [y7]—745.420+ |
| EDPLLTDALAGQR | 699.865++ | 25 | L [y9]—944.516+ |
| | | | T [y8]—831.432+ |
| | | | D [y7]—730.384+ |
| **RthD** | | | |
| LIDEEPYR | 517.761++ | 18 | D [y6]—808.347+ |
| | | | E [y5]—693.320+ |
| | | | E [y4]—564.278+ |
| ATGLSDEEFLAR | 654.825++ | 23 | S [y8]—966.453+ |
| | | | D [y7]—879.421+ |
| | | | E [y6]—764.394+ |
| **RthE** | | | |
| IPVYLAALGPK | 571.353++ | 20 | V [y9]—931.561+ |
| | | | Y [y8]—832.493+ |
| | | | L [y7]—669.430+ |
| IDVGSAVLQIPAR | 669.891++ | 24 | A [y8]—867.541+ |
| | | | V [y7]—796.504+ |
| | | | L [y6]—697.436+ |
| **RthI** | | | |
| GQLPEGAWR | 507.262++ | 18 | L [y7]—828.436+ |
| | | | P [y6]—715.352+ |
| | | | E [y5]—618.299+ |
| LGTADLWLR | 522.795++ | 18 | A [y6]—773.430+ |
| | | | D [y5]—702.393+ |
| | | | L [y4]—587.366+ |
| **RthJ** | | | |
| LYGGAATDIPHVR | 685.365++ | 24 | A [y8]—908.495+ |
| | | | T [y7]—837.458+ |
| | | | D [y6]—736.410+ |
| SELAGVFADLLR | 645.856++ | 23 | G [y8]—890.509+ |
| | | | V [y7]—833.488+ |
| | | | F [y6]—734.420+ |

[a]CE, collision energy.

1.1.383) (76) using R (version 4.0.5) (77) for Welch's *t* test and Tukey's *post hoc* test. Visualization of the data was done using the ggplot2 package (78).

**Data availability.** The MRM data have been deposited in PeptideAtlas (79) with the data set identifier PASS01674.

## SUPPLEMENTAL MATERIAL

Supplemental material is available online only.

**SUPPLEMENTAL FILE 1**, PDF file, 1.1 MB.

## ACKNOWLEDGMENTS

B.D. is supported by grants from the Ghent University research council supporting a proteomics expertise center. We acknowledge the Hercules initiative for the multispectral imaging platform that was granted (grant number AUGE/15/17).

We are thankful to Isolde Francis at California State University—Bakersfield for providing cosmid 2012 and helpful discussions.

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
