## [Reviewer comments · Microbiology Spectrum]

**Microbiology
Spectrum**

Identification of Novel Rotihibin Analogues in *Streptomyces scabies* including Discovery of its Biosynthetic Gene Cluster

Soren Planckaert, Benoit Deflandre, Anne-Mare de Vries, Maarten Ameye, José C. Martins, Kris Audenaert, Sébastien Rigali, and Bart Devreese

Corresponding Author(s): Bart Devreese, Ghent University

Review Timeline:

Submission Date:

June 28, 2021

Accepted:

June 29, 2021

Editor: Jeffrey Gralnick

Reviewer(s): The reviewers have opted to remain anonymous.

Transaction Report:

DOI: <https://doi.org/10.1128/Spectrum.00571-21>

June 29, 2021

Prof. Bart Devreese
Ghent University
Biochemistry and Microbiology, Faculty of Sciences
K.L. Ledeganckstraat 35
Ghent 9000
Belgium

Re: Spectrum00571-21 (Identification of Novel Rotihibin Analogues in *Streptomyces scabies* including Discovery of its Biosynthetic Gene Cluster)

Dear Prof. Bart Devreese:

Based on your thoughtful responses to the prior round of review and in consultation with another Editor, your manuscript has been accepted, and I am forwarding it to the ASM Journals Department for publication. You will be notified when your proofs are ready to be viewed. We will require the addition of a 'Data Availability' heading in the Methods section where you describe where the proteomic data can be accessed, but this should be easily done by our editorial staff.

Sincerely,

Jeffrey Gralnick
Editor, Microbiology Spectrum
